# PET-MAD as a lightweight universal interatomic potential for advanced materials modeling

Arslan Mazitov [1,2] ✉, Filippo Bigi[1,2], Matthias Kellner[1], Paolo Pegolo [1], Davide Tisi [1], Guillaume Fraux[1], Sergey Pozdnyakov[1], Philip Loche [1] & Michele Ceriotti [1] ✉

Machine-learning interatomic potentials have greatly extended the reach of atomic-scale simulations, offering the accuracy of first-principles calculations at a fraction of the cost. Leveraging large quantum mechanical databases and expressive architectures, recent universal models deliver qualitative accuracy across the periodic table but are often biased toward low-energy configurations. We introduce PET-MAD, a generally applicable interatomic potential trained on a dataset combining stable inorganic and organic solids, systematically modified to enhance atomic diversity. Using a moderate but thoroughly consistent level of electronic-structure theory, we assess PET-MAD's accuracy on established benchmarks and advanced simulations of six materials. Despite the small training set and lightweight architecture, PET-MAD is competitive with the state-of-the-art machine-learned interatomic potentials for inorganic solids, while also being reliable for molecules, organic materials, and surfaces. It is stable and fast, enabling the near-quantitative study of thermal and quantum mechanical fluctuations, functional properties, and phase transitions out of the box. It can be efficiently fine-tuned to deliver full quantum mechanical accuracy with a minimal number of targeted calculations.

Efficient methods for solving the electronic structure problem for molecules and materials have made it possible to predict their structure, stability, and properties from first principles. However, the associated computational cost and poor scaling with system size severely limit the complexity, time, and length scale accessible to simulations. Machine-learning (ML) models that aim to replace first-principles methods to evaluate energy and forces—so-called machine-learning interatomic potentials (MLIPs)—address this limitation by training on a set of reference quantum mechanical calculations, and then making accurate yet inexpensive ML predictions, for both materials[1–3] and (bio)molecules[4–7].

For almost two decades, this scheme has been used successfully to study a large number of chemical systems for which first-principles calculations are too expensive, and empirical force fields are either not available or not sufficiently accurate[8–10]. Nonetheless, the aforementioned approach typically implies parameterizing a specific MLIP for every new system, which requires a considerable number of ab initio calculations, as well as substantial human effort and expertise in fitting the potential. In contrast to this family of single-purpose potentials, recent years have seen the development of several general-purpose, or universal models[11–13], which aim to be applicable to a large range of distinct chemical systems, either out of the box or after minimal fine-

[1]Laboratory of Computational Science and Modeling, Institut des Matériaux, École Polytechnique Fédérale de Lausanne, Lausanne, Switzerland. [2]These authors contributed equally: Arslan Mazitov, Filippo Bigi. ✉e-mail: arslan.mazitov@epfl.ch; michele.ceriotti@epfl.ch

tuning. These models provide a qualitative description of the atomic-scale interactions across the periodic table, although their level of accuracy for a specific application depends on how closely the problem of interest aligns with the considerations underlying the construction of the reference dataset. For instance, many existing datasets are built to represent a collection of stable materials, and to support the search for new materials with optimal properties[14–17]. To complicate things, the accuracy of general-purpose potentials is often assessed using inconsistent levels of ab initio theory, making it difficult to isolate the shortcomings of the model from the discrepancy in the electronic-structure reference. In this work, we use the Massive Atomistic Diversity (MAD) dataset, which incorporates a high degree of chemical and structural diversity while employing highly converged, internally consistent reference calculations. This allows for a quantitative assessment of the performance of fitted models across the periodic table, since MAD's ab initio settings are broadly applicable to most chemical systems, although the comparatively simple exchange-correlation functional we use may not always be equally accurate. Energies and forces are fitted using a Point Edge Transformer (PET) graph neural network[18], resulting in a general-purpose MLIP, which we name PET-MAD. We evaluate the accuracy of the resulting model on a wide range of public datasets, ensuring consistency in the electronic-structure settings, and showing that PET-MAD achieves competitive performance on several external benchmarks despite being trained on 2-3 orders of magnitude fewer structures. We then critically assess the performance of PET-MAD in highly non-trivial atomistic simulations, including accelerated statistical sampling, quantum nuclear fluctuations, and predicting functional properties. We focus on six examples, motivated by their scientific interest, diversity in material classes and physical effects being probed, as well as the fact that each has previously been studied using ad hoc MLIPs—namely, lithium thiophosphate, gallium arsenide, a CoCrFeMnNi high-entropy alloy, liquid water, succinic acid, and barium titanate. For each example, we quantitatively assess the accuracy of PET-MAD by comparing it against both a bespoke model trained with compatible energetics, which provides an extremely close match to the electronic-structure reference, as well as a fine-tuned model that enables PET-MAD to achieve the same level of accuracy with a small number of additional training structures. PET-MAD demonstrates that accurate, fast, and robust universal models can be trained using a tiny fraction of the structures of last-generation datasets, and provides a feature-rich framework for advanced atomistic simulations, which includes direct-force acceleration and inexpensive end-to-end uncertainty quantification.

## Results

PET-MAD is a generally applicable machine-learning interatomic potential based on PET (an unconstrained, transformer-based graph neural network architecture) and a custom training set built on the principles of Massive Atomistic Diversity (MAD) and internal consistency of the reference energetics. We thoroughly test the performance of PET-MAD against several benchmark datasets and compare it with four widely used universal potentials. Six case studies complete our analysis, showcasing and quantitatively assessing PET-MAD for diverse classes of materials and advanced atomistic simulations, comparing a system-specific PET version trained on an ad hoc dataset, the generally-applicable PET-MAD, and the versions fine-tuned to each system. We emphasize that in all of these examples, we focus on the consistency of the model with the electronic-structure reference, irrespective of the actual accuracy of the latter in reproducing experimental quantities.

### MAD dataset

Most of the existing efforts to generate datasets to train universal models focus on either inorganic crystals[14,16,19] or molecular compounds[20], and aim to include as many structures as possible. PET-

MAD is trained on the Massive Atomistic Diversity (MAD) dataset, which is based on a different philosophy. First, it aims to push the limits of universality by including both organic and inorganic atomistic systems of all possible dimensionalities. This approach has already been shown to deliver good transferability upon including up to 45 elements in the training data[21]. Second, our resulting model is meant to work in complex atomistic simulation protocols describing a wide range of thermodynamic conditions, which requires covering a large configuration space and being computationally affordable. To this end, the MAD dataset includes randomized and highly-distorted structures in the training set, applying to solids and surfaces ideas similar to those that drove the development of "mindless" molecular benchmark sets[22]. Third, the reference electronic-structure calculations are designed to be robust and consistent, to ensure that the structural and chemical motifs are treated in the same way, regardless of the type of structure they are part of, which is important for a coherent structure-energy mapping. This choice neglects, or approximates poorly, the description of effects such as spin polarization, electron correlation, and dispersion, which are important for certain classes of materials but cannot be applied consistently across the MAD dataset. Details of the electronic-structure calculations are given in "First-principles calculations". Last but not least, we restrict the dataset to fewer than 100,000 structures in order to reduce the time and cost of training, making the entire training procedure accessible to a wider community. As we shall see, the strategy we use allows us to preserve the representative power of the dataset.

The MAD dataset contains 95,595 structures of 85 elements in total (with atomic numbers ranging from 1 to 86, except Astatine) and consists of 8 subsets that cover a diverse range of chemical and structural motifs. The largest subset, MC3D, contains 33,596 bulk crystal structures from the Materials Cloud 3D database[23]. Its diversified variants, MC3D-rattled and MC3D-random, comprise 30,044 and 2800 structures, respectively, where the former is generated by adding Gaussian noise to atomic positions, and the latter by randomly reassigning atomic species within selected crystals. Additional structural diversity is introduced by MC3D-surface, which includes 5589 surface slabs cleaved along low-index planes, and MC3D-cluster, consisting of 9071 nanoclusters cut from random bulk environments. Complementing these, MC2D provides 2676 two-dimensional crystals from the Materials Cloud 2D database[24]. Finally, the SHIFTML-molcrys and SHIFTML-molfrags subsets contribute 8578 molecular crystals and 3241 neutral molecular fragments, respectively, both derived from the SHIFTML and Cambridge Structural Database collections[25–27]. A more detailed description of the subsets is given in Methods, and in Supplementary Table 1.

The number of structures in each subset represents the total number of structures that were successfully converged in DFT calculations, after filtering out a few outlier configurations with forces above a large threshold (100 eV/Å for MC3D-rattled and MC3D-random, and 10 eV/Å for the other subsets). Specific details on the generation of MAD subsets are given in "MAD dataset construction" and in a dedicated public data record (see "Data availability").

### Model architecture and training

The PET architecture[18] is a rotationally unconstrained and transformer-based graph neural network (GNN) which has a high descriptive power (as every transformer layer can be scaled to be a universal approximator)[18] and low inference cost (as we shall demonstrate in "Benchmarking"). For PET-MAD, we chose the architecture based on an extensive hyperparameter search, resulting in a model with approximately 2.8 million parameters in total. Before training, each subset of MAD was randomly shuffled and further split into training, validation, and test subsets in fractions of 80%, 10%, and 10%, respectively. All subsets were then merged to obtain the final training, validation, and test sets. More details on the architecture and training are available in

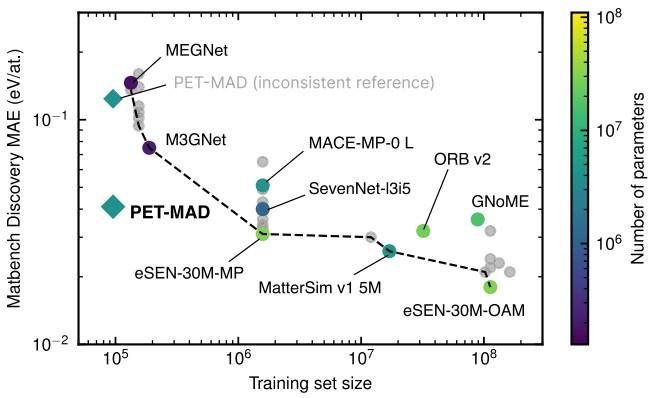

**Fig. 1 | Pareto frontier of various universal MLIPs.** The number of configurations in the training set is shown on the x-axis, and y-axis shows mean absolute error (MAE) for the energy-above-hull property on a subset of the WBM dataset[15] (a part of the Matbench Discovery benchmark[33]). Colored dots represent a set of selected models, each with a corresponding number of parameters shown in color, while other models from the benchmark are shown with grey dots. PET-MAD results are shown with diamonds for both consistent and inconsistent DFT references, where the latter is shown with a shaded label. PET-MAD significantly pushes the frontier (dotted line) downward, achieving the accuracy of other advanced universal MLIPs like SevenNet and GNoME using 1-3 orders of magnitude less data.

"PET model", "Training of PET-MAD", and Supplementary Information. In order to improve the performance of PET-MAD on specific chemical systems, we used a fine-tuning strategy based on the low-rank adaptation technique (LoRA)[28]. Details on fine-tuning are further discussed in "Fine-tuning".

The model also includes uncertainty-quantification capabilities, based on the last-layer prediction rigidity (LLPR) method[29]. Error estimation is especially important for a model aiming to be generally-applicable, and that (despite the high diversity of the MAD training set) may often be used for out-of-domain predictions. As shown in Supplementary Note 8, the calibrated LLPR uncertainties correlate well with the empirical test-set errors, while adding negligible overhead on top of energy predictions. Given that PET-MAD is intended for use in advanced simulations that estimate observable quantities through sophisticated sampling protocols, it is essential to be able to propagate the energy errors onto the final quantity of interest. To this end, we generate a shallow ensemble based on the LLPR covariance[29]. Unlike standard ensemble-based methods[30], which typically introduce a computational overhead proportional to the number of ensemble members, both LLPR and the shallow ensemble method[31] allow uncertainty quantification at negligible additional cost. Moreover, they allow one to compute errors on any derived quantities using direct ensemble propagation, as we demonstrate for free-energy calculations in "Melting point of GaAs", and for phonon dispersion curves in Supplementary Note 9. Details on our uncertainty-quantification approach are summarized in "Uncertainty quantification" and explained thoroughly in the Supplementary Information. Even though PET-MAD is designed to evaluate forces using backpropagation, we also train a separate head that predicts forces directly as a function of the atomic coordinates. Avoiding backpropagation makes the model two to three times faster (as we show in "Benchmarking"), but should be used with care, as it violates energy conservation with pathological consequences on sampling accuracy[32]. We address these problems in Supplementary Note 12, where we also show how a multiple-time-step integrator allows one to retain the computational advantages of direct force predictions without introducing sampling artifacts.

## Benchmarking
First, we demonstrate the efficacy of combining the MAD dataset and the PET architecture by comparing the accuracy of PET-MAD against

other universal models on the popular Matbench Discovery benchmark[33]. We analyze the overall gain in models' accuracies with the increase in the training set size and the number of trainable parameters.

Since PET-MAD is trained on the MAD dataset, which has different DFT settings compared to the Matbench Discovery (which shares common settings with the training sets of other reference models), we recomputed a random subset of the benchmark containing 555 structures (excluding lanthanides and actinides) using MAD-compatible DFT settings. We evaluate the accuracies of all models on this subset, picking for each model the DFT reference compatible with its training set. Fig. 1 demonstrates the Pareto frontier of the models' data efficiency: Pareto-optimal models efficiently use additional training data to increase their accuracy. PET-MAD significantly improves on existing models, achieving the accuracy of other advanced universal MLIPs such as SevenNet and GNoME using 1–3 orders of magnitude less data. For illustrative purposes, the figure also shows the accuracy of PET-MAD evaluated against the inconsistent Matbench Discovery energetics: the 3 × larger error is entirely due to the discrepancy in the reference data. A thorough analysis of DFT consistency effects is discussed in the Supplementary Note 4.

The figure also shows that many of the optimal models use a very large number of parameters, which makes them accurate but computationally demanding, and therefore not suitable for the long, complex simulation tasks that PET-MAD is designed for. For this reason, we will focus on medium-size models for the rest of these benchmarks: MACE-MP-0[11], MatterSim[12], Orb-v2[13] and SevenNet[34]. Like PET-MAD, these models have been designed and are widely used for advanced materials simulations beyond single-point calculations. We extend the comparison on Matbench Discovery to include other popular atomistic machine learning benchmarks for bulk inorganic systems, molecular systems, and catalytic applications, namely MPtrj[14], Alexandria[19], OC2020 (S2EF)[35], SPICE[20], and MD22[36]. A detailed description of the structure types entering each benchmark is provided in Supplementary Table 2. In evaluating the benchmarks, it is important to keep in mind that (1) most of the reference models are larger than PET-MAD (we use the versions with the largest number of parameters in their families, MACE-MP-0-L and MatterSim-5M, Orb-v2, SevenNet-l3i5 having respectively about 15.8 M, 4.6 M, 25 M, and 1.17 M parameters, as opposed to 2.8 M for PET-MAD) and (2) they are trained on much larger datasets (1.58 M, 17 M, 32.1 M, and 1.58 M structures, respectively).

The benchmarking results are presented in Table 1. We find that PET-MAD achieves high accuracy in predicting both energies and interatomic forces, outperforming MACE-MP-0-L in most cases, and competing closely with MatterSim-5M and SevenNet-l3i5 even for the inorganic datasets they are trained on. Upon using a consistent level of DFT theory, PET-MAD outperforms all other models in predicting raw energies of the WBM crystals from the Matbench Discovery benchmark[33]. Orb-v2 performs significantly better than all other models on MPtrj and Alexandria, which are part of its training set. PET-MAD outperforms all other models on molecular datasets such as SPICE and MD22, and matches Orb-v2 also on the OC2020 S2EF – which is a strongly extrapolative exercise as MAD does not include configurations of adsorbed molecules.

The reference models perform much worse than PET-MAD on the (consistently recomputed) MAD benchmark. This is not too surprising, as MAD is designed to be more diverse than the datasets on which these models are trained. Still, it is instructive to assess separately the accuracy of the various models on the different sections of MAD (Fig. 2). PET-MAD outperforms MACE-MP-0-L, MatterSim-5M, and SevenNet-l3i5 in almost all cases. The accuracy is similar (and in a few cases slightly better) for the stable inorganic structures, including bulk (MC3D) and layered (MC2D) compounds. MC3D-derived surfaces and clusters show slightly larger force and considerably larger energy

**Table 1 | Comparison of PET-MAD accuracies on popular atomistic machine learning datasets against MACE-MP-0-L, MatterSim-5M, Orb-v2, and SevenNet-l3i5 models**

| Dataset | PET-MAD | MACE-MP-0-L | MatterSim-5M | Orb-v2 | SevenNet |
|---|---|---|---|---|---|
| MAD | **17.6** \| **65.1** | 81.6\|181.5 | 47.3\|133.7 | 52.9\|96.2 | 82.1\|173.5 |
| MPtrj | 22.3\|77.9 | 15.1\|50.8 | 21.3\|61.4 | **5.6** \| **21.9** | 9.8\|25.5 |
| Matbench | **31.3** \| — | 58.5\|— | 38.2 \|— | 37.9\|— | 47.5\|— |
| Alexandria | 49.0\|66.8 | 65.4\|79.5 | 21.2\|39.9 | **13.2** \| **10.5** | 47.6\|70.3 |
| OC2020 | **18.3** \|114.5 | 82.4\|169.6 | 31.5\|119.2 | 19.8\| **99.3** | 45.7\|162.7 |
| SPICE | **3.7** \| **59.5** | 10.6\|166.8 | 21.3\|145.6 | 59.0\|140.8 | 11.3\|139.1 |
| MD22 | **1.9** \| **65.6** | 9.4\|182.9 | 28.6\|160.4 | 174.3\|220.7 | 11.1\|146.2 |

For each dataset and model, mean absolute errors are reported for raw energy|forces predictions (in units of meV/atom|meV/Å). The best results for each benchmark are shown in bold. We do not show force errors for the Matbench Discovery subset, since forces are not available in the reference data.

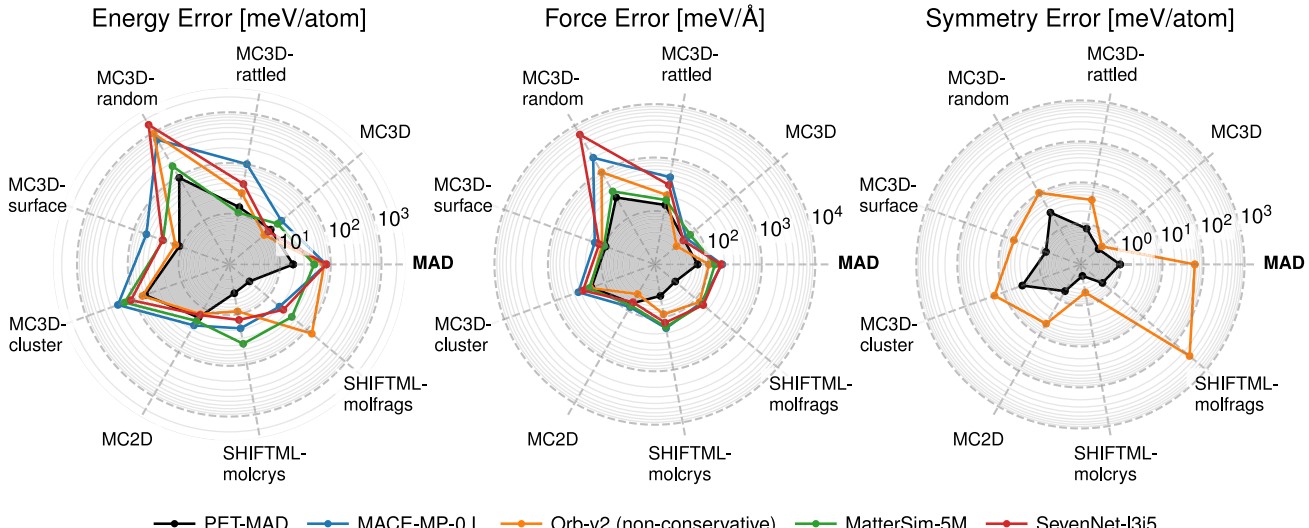

**Fig. 2 | Comparison of the accuracies of the universal MLIPs on the MAD dataset.** PET-MAD (black line), MACE-MP-0-L (blue line), Orb-v2 (orange line), MatterSim-5M (green line), and SevenNet-l3i5 (red line) models' results are resolved by different MAD subsets. The left and center panels show the mean absolute errors in energies and force predictions. The right panel shows the rotational discrepancy in energy predictions of the unconstrained models (PET-MAD and Orb-v2). Results for all models except PET-MAD were obtained using consistently recalculated reference values (i.e., computed with MPtrj dataset settings) to achieve consistency with the training sets of the corresponding reference models. See also Table 1 for benchmarks on out-of-sample datasets.

errors. Errors of the other universal models are up to 50 times larger for the distorted subsets (MC3D-rattled and MC3D-randcomp) which contain especially unusual, diverse configurations. MatterSim-5M, which is trained on a broader, yet unpublished, set of configurations, performs substantially better than the other models. The most notable difference occurs in the molecular systems (SHIFTML-molcrys and SHIFTML-molfrags), where PET-MAD dramatically outperforms all other models. This result is again expected, as organic systems represent a completely different region of configuration space, which is heavily undersampled in the case of inorganic crystal datasets. The last model from the reference list, Orb-v2, requires a separate discussion, as it provides by default direct, non-conservative force predictions. Orb-v2 outperforms all other models (including, marginally, PET-MAD) on relaxed inorganic systems, presumably thanks to the combination of a flexible, unconstrained architecture and the large training set (which includes both the MPtrj[14] and Alexandria[19] datasets). However, it still provides worse accuracy on rattled, random composition structures, surfaces and molecular systems compared to PET-MAD. Both PET-MAD and Orb-v2 use architectures that do not enforce exact rotational equivariance, which is learned approximately by data augmentation. The resulting symmetry breaking can be monitored and controlled easily[37], but it is important to assess its extent. To do so, we estimated the symmetry error in PET-MAD and Orb-v2 energy

predictions by applying a series of rotations on a Lebedev-Laikov grid of order 9 for each structure in the MAD subset and calculating the standard deviation in the predictions (Fig. 2, right panel). In most cases, the rotational discrepancy of the PET-MAD model is one or two orders of magnitude smaller than the corresponding prediction error, below 1 meV/atom for all subsets except MC3D-random and MC3D-cluster. Orb-v2 shows significantly higher discrepancies in energy predictions, which are at times comparable to the actual errors.

The PET-MAD model is also computationally efficient, which facilitates simulating larger time and length scales. We compared the throughput of molecular dynamics simulations of systems of different sizes and densities (solid aluminum, diamond, and liquid water, see Fig. 3) and, when available, used lighter versions of the reference models compared to those used in the accuracy benchmarks (namely, MACE-MP-0 M, MatterSim-1M, and SevenNet-0). These models have fewer parameters and thus allow faster inference at the expense of accuracy, making this a more challenging test for the speed of PET-MAD. Nevertheless, we see that PET-MAD is faster and more memory efficient than all models. When exploiting the computational advantage of direct-force prediction, PET-MAD is also competitive with the non-conservative Orb-v2 model.

The takeaway is that an unconstrained architecture and a problem-agnostic construction of the training set make PET-MAD

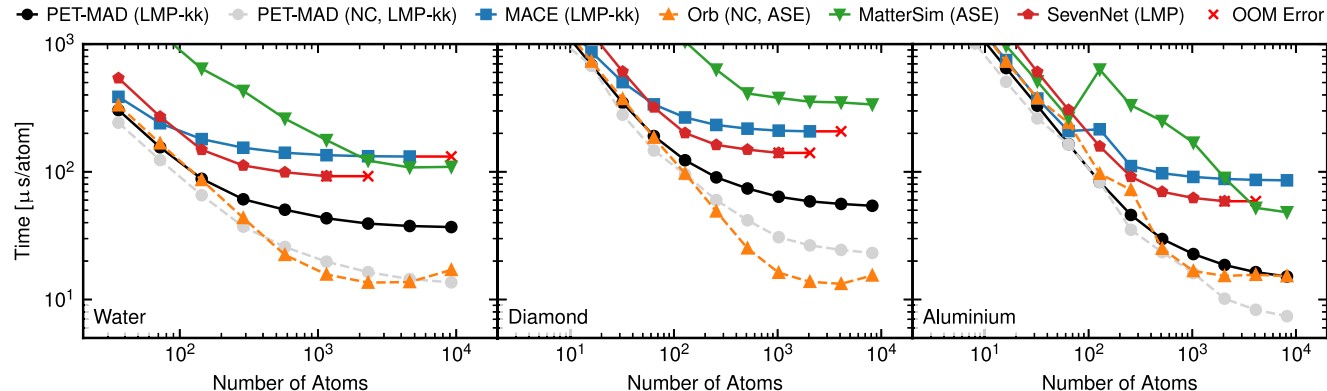

**Fig. 3 | Inference time of several universal MLIPs evaluated over different bulk materials and system sizes.** For each MLIP, we use its LAMMPS[65] interface if available, preferably choosing the Kokkos-enabled[67] (kk) version, or its ASE[68] interface otherwise. All timings were measured on a single NVIDIA H100 GPU. Missing points for MACE-MP-0 and SevenNet caused by out-of-memory errors are marked with red crosses. Color code of the lines is consistent with the one used in Fig. 2. The non-conservative (NC) versions of Orb-v2 and PET-MAD are shown in dashed orange and grey lines, respectively. These models benefit from a theoretical speedup, but can violate conservation of energy, often resulting in ill-behaved molecular dynamics[32]. The model versions used were MatterSim-v1.0.0-1M, MACE-MP-0 (M), Orb-v2, SevenNet-0 (11Jul2024).

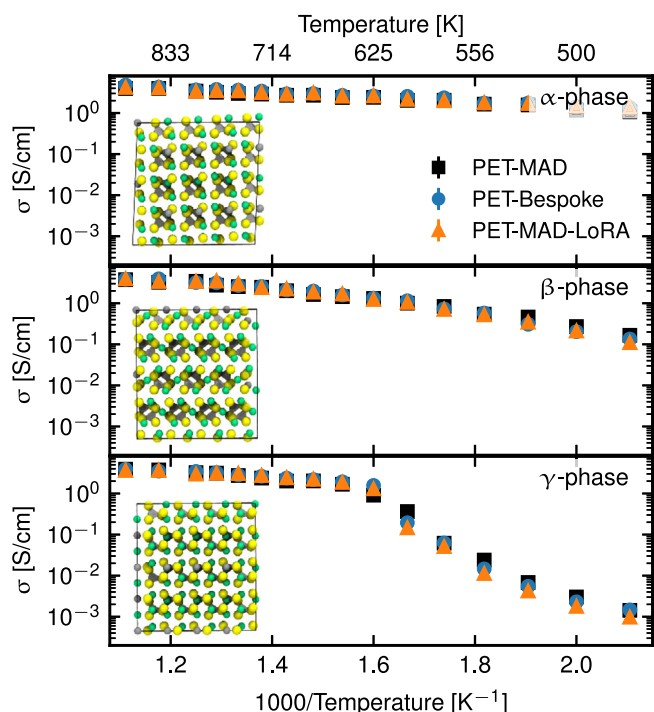

**Fig. 4 | Temperature dependence of the ionic conductivity $\sigma$ for the $\alpha$, $\beta$, and $\gamma$ phases of Li$_3$PS$_4$.** The figure compares the values of $\sigma$ from PET-MAD (black), a fine-tuned model (orange), and a bespoke PET model trained from scratch (blue).

competitive in speed and accuracy with models that have a larger number of parameters and are trained on much larger datasets (Fig. 1). This has important implications to guide the design of future datasets and provides a more sustainable and accessible platform to support model development.

Benchmarking accuracies alone, however, do not ensure that PET-MAD can be reliably used in realistic atomistic modeling scenarios. Therefore, in the following sections, we demonstrate the applicability of PET-MAD for atomistic simulations by comparing its performance against single-purpose and fine-tuned models for six diverse and challenging use cases.

## Ionic transport in lithium thiophosphate

Lithium thiophosphates (LPS) are a class of materials that have been intensely studied as electrolytes for solid-state batteries[38]. The archetypal member of this class, Li$_3$PS$_4$, has been the subject of several computational investigations, including ref. 39, which we use both as a blueprint for this benchmark and the source of the system-specific dataset. Following the same approach as in that reference, we compute the ionic conductivity $\sigma$ (one of the key properties for the technological applications of LPS) of three known phases of Li$_3$PS$_4$, $\alpha$, $\beta$ and $\gamma$[40], using molecular dynamics and the Green-Kubo theory of linear response[41].

We compare the results obtained using the PET-MAD potential with a single-purpose PET potential trained from scratch over the dataset from ref. 39 (PET-Bespoke), and with one fine-tuned over the same dataset using the LoRA technique (see "Fine-tuning" for more details, and the SI for a full discussion). For this dataset, the validation mean absolute error (MAE) of PET-MAD for energy|forces is 4.9 meV/atom|63.9 meV/Å, to be compared with 1.2 meV/atom|35.6 meV/Å, for a model trained from scratch on the LPS dataset, and 1.3 meV/atom|36.0 meV/Å for the fine-tuned model. Fig. 4 shows the values of the conductivity for all the phases and over a wide range of temperatures computed with PET-MAD (black), a model fine-tuned using the LoRA algorithm with rank 8 (orange), and the PET-Bespoke model (blue). For all three phases, PET-MAD is in excellent agreement with the bespoke and fine-tuned models, demonstrating quantitative accuracy despite the larger validation error. For the $\gamma$-phase, PET-MAD slightly overestimates the temperature at which the transition to a high-conductivity phase occurs. Except for this minor discrepancy, PET-MAD shows excellent agreement with the more accurate dedicated models, capturing the change of behavior in $\sigma$ following the rotation of the PS$_4$ tetrahedra[39]. The results agree quantitatively with those of ref. 39, despite being based on slightly different DFT parameters and relying on a different class of MLIPs.

## Melting point of GaAs

Gallium Arsenide (GaAs) is a III-V semiconductor whose properties make it a good choice to manufacture the high-quality optical and electrical devices, as well as high-end photovoltaics.

The growth of GaAs nanostructures often relies on the coexistence of solid GaAs and molten Ga in an As-rich atmosphere[42]. Corresponding computational studies typically require MLIPs that can accurately describe different phases and compositions across the Ga/

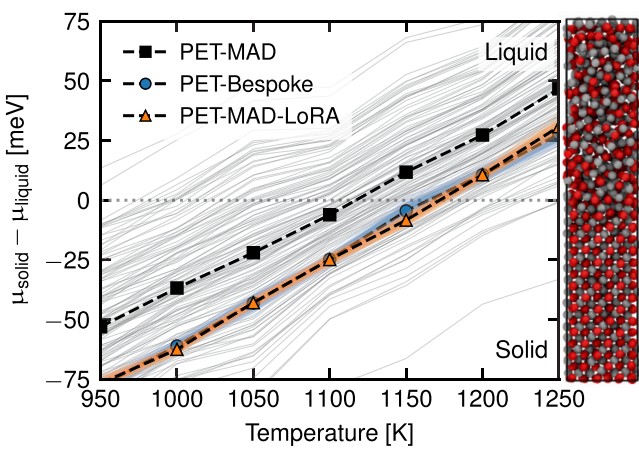

**Fig. 5 | Chemical potential differences between liquid and solid GaAs phases as a function of temperature.** The results are computed with PET-MAD (black), PET-Bespoke (blue) and PET-MAD-LoRA (orange) models. The dashed solid lines indicate the mean values of the predicted $\Delta\mu$, and the light colored lines represent the reweighted predictions from the individual members of the shallow ensemble. The large spread of the PET-MAD ensemble predictions correctly reflects the discrepancy with respect to the Bespoke and LoRA-fine-tuned model.

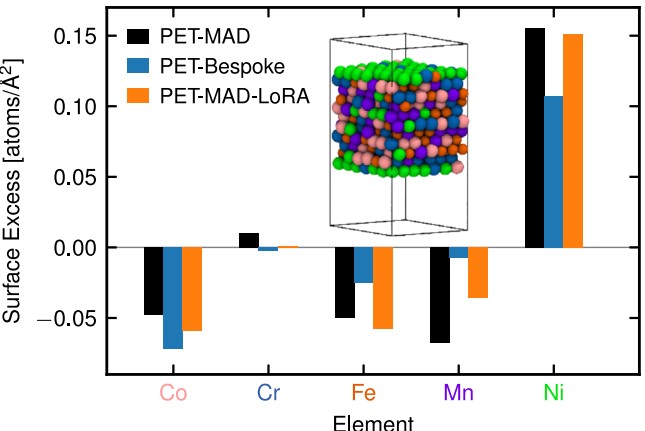

**Fig. 6 | Gibbs surface excess $\Gamma_a$ of the elements on a (111) surface of the CoCrFeMnNi alloy.** The results are obtained from the REMD/MC simulation at 800 K performed with the pre-trained PET-MAD model (black bars), bespoke PET model trained on a subset of the HEA25S dataset (blue bars), and the PET-MAD-LoRA model, fine-tuned on the same subset (orange bars). Both pre-trained and fine-tuned models result in almost identical segregation patterns with the surface enriched in nickel, which is similar to one from ref. 48. The inconsistency between the bespoke and fine-tuned models is most likely due to the bespoke model's overfitting.

As phase diagram. Electronic properties change wildly between different phases, posing additional challenges to the construction of empirical potentials. Following the work of Imbalzano et al.[43], we repeat the calculation of the melting point of stoichiometric GaAs at ambient pressure with the interface pinning method. This method computes the differences in chemical potentials between the liquid and solid phases, and the melting point is determined by identifying the temperature at which the chemical potential difference becomes zero. A combination of the LLPR ensemble predictions with the reweighting technique allows us to propagate errors on the chemical potential curves[44], and therefore assess the contribution of the epistemic part of the error to the melting point error caused by limitations of the ML fit.

Figure 5 demonstrates these calculations, comparing the PET-MAD model with a bespoke PET model trained on the reference GaAs training set, and the LoRA-fine-tuned model on the same training set. The test set errors (MAE) on energy|forces are 14.4 meV/at.|74.1 meV/Å, for PET-MAD, 0.7 meV/at.|29.0 meV/Å, for PET-Bespoke and 1.3 meV/at.|45.3 meV/Å, for PET-MAD-LoRA. When it comes to the estimation of the melting point, the LoRA and Bespoke models are in quantitative agreement (1169 ± 3 K vs 1169 ± 4 K) whilst the PET-MAD model predicts a slightly smaller value of 1111 ± 72 K. The predicted error is consistent with the lower accuracy of the general-purpose PET-MAD, which highlights the importance of the inexpensive uncertainty propagation afforded by the LLPR ensemble architecture implemented in our PET models. It should be noted that the error of the general-purpose model against the bespoke potentials is very small compared to the deviation of the computed melting point from the experimentally measured melting point (1511 K), and with the typical errors of empirical force fields. This large discrepancy is consistent with that observed in ref. 43, and can be attributed to shortcomings of the reference electronic-structure calculations – which is not uncommon as the computed melting point of a given material can vary by hundreds of Kelvins depending on the choice of DFT functional[45].

### Surface segregation in high-entropy alloys

High-entropy alloys – containing 5 or more metals in roughly equimolar composition – often possess excellent mechanical properties[46], and perform well as heterogeneous catalysts[47]. Investigating their properties and exploring efficiently their composition space requires

versatile models that can handle a high degree of chemical diversity. We replicate some of the simulations in ref. 48, using PET-MAD to study the segregation of different elements at the (111) surface of the CoCrFeMnNi alloy, a prototypical HEA[46]. Modeling of the differential surface propensity of the elements in the alloy – which is central to the applications of HEAs as heterogeneous catalysts – requires a combination of replica-exchange molecular dynamics with Monte-Carlo atom swaps (REMD/MC) to overcome the slow diffusivity, and the existence of free-energy barriers to segregation starting from a random alloy. This example allows us to demonstrate the capabilities of PET-MAD in complex computational workflows, to assess the ability of the model to describe surface effects that are often overlooked in other universal models[49], and to test finetuning over a dataset with a greater degree of chemical diversity than the other examples we consider. We use the Gibbs surface excess calculated at 800 K using the REMD/MC protocol (see Methods for computational details) as a measure of surface segregation in a CoCrFeMnNi surface slab. We compare three models: the pre-trained PET-MAD, PET trained from scratch (PET-Bespoke), and the fine-tuned model (PET-MAD-LoRA), where the last two were trained on a subset of the HEA25S dataset[48], recomputed with MAD dataset DFT settings for consistency (details on the subset selection are given in Methods). The validations MAEs of all three models on energy|forces are 25.8 meV/atom|175.1 meV/Å, 14.6 meV/atom|138.3 meV/Å, and 9.4 meV/atom|124.8 meV/Å, respectively.

Results of the surface excess calculation are given in Fig. 6. First, we note that the segregation pattern obtained from both pre-trained and fine-tuned models is almost identical and quantitatively matches the results from ref. 48, which corresponds to surface layers enriched in nickel and depleted in other elements. Both these models agree qualitatively with the results of the PET-Bespoke model, but there is a more pronounced quantitative difference in the surface excess values. In interpreting this discrepancy, one should keep in mind that, contrary to other examples, we only recomputed about 2000 structures out of the 30'000 in the HEA25S, and so a bespoke model incurs a high risk of overfitting – which is also consistent with the learning curves reported in the SI. This case study demonstrates that pre-trained PET-MAD is capable of capturing all effects relevant to surface segregation in HEAs and providing advanced sampling capabilities of the HEA

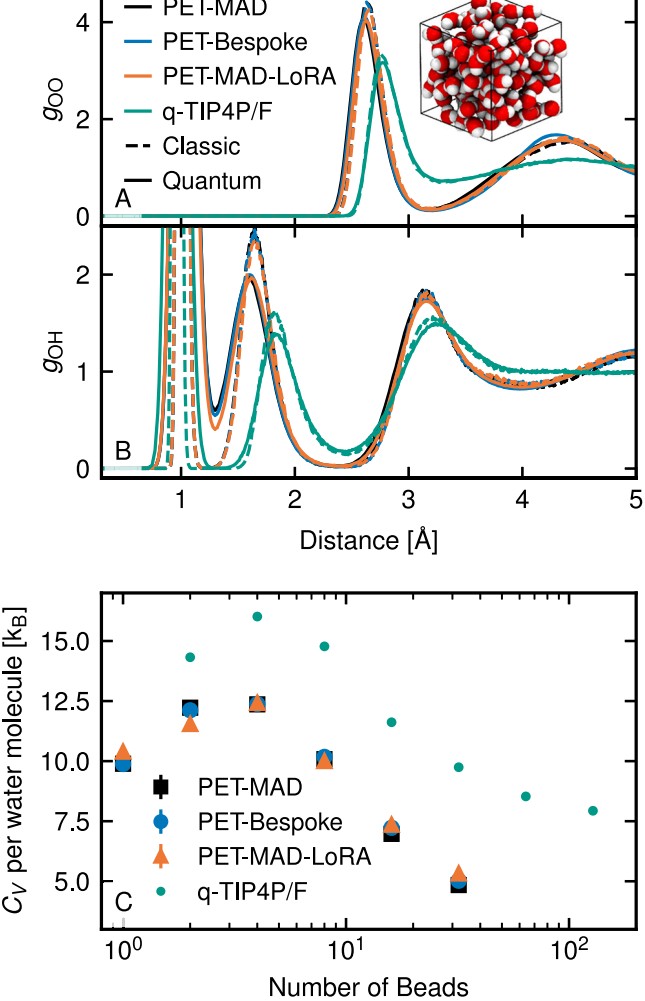

**Fig. 7 | Radial distribution functions $g$ (top) and heat capacity $C_V$ (bottom) of liquid water with nuclear quantum effects.** The results are obtained using PIMD simulations at 298 K and 1 atm. with PET-MAD, a fine-tuned version of PET-MAD, and a bespoke PET model, shown in black (squares), orange (triangles), and blue (circles) lines and symbols, respectively. Solid lines show the radial distribution function of O-O and O-H pairs from the PIMD (quantum) simulation, and classic MD results are shown in dashed lines for comparison. $C_V$ is given as a function of the number of beads employed in the PIMD simulation. Results on the $g$ and $C_V$ obtained with the empirical q-TIP4P/F forcefield (reproduced from ref. 69) are also included for comparison and shown in green lines and dots.

chemical and structural space without being specifically trained on HEA data.

## Quantum nuclear effects in liquid water

Water is a widely studied system because of its high biological, environmental, and technological significance, as well as the many anomalies in its physical properties, which can be traced to the strong hydrogen bonds that form local tetrahedral motifs. Due to its light hydrogen atoms, accounting for nuclear quantum effects (NQEs) is fundamental in order to extract accurate observables for liquid water, even at room temperature[50]. In this example, we employ path integral molecular dynamics (PIMD) to model nuclear quantum effects in the simulation of a medium-sized periodic system consisting of 128 water molecules (384 atoms).

We first evaluate the O-O and O-H pair correlation functions, which report on inter-molecular structural correlations, and show the extent of intra-molecular quantum fluctuations, respectively.

Additionally, we calculate the constant-volume heat capacity at 298 K – a physical quantity that exhibits very pronounced quantum effects, and that requires the use of sophisticated path integral estimators that are notoriously difficult to converge[51]. We compare (1) the PET-MAD model, (2) a system-specific PET model trained on the dataset from ref. 52, with energies and forces recomputed at the same level of theory used for MAD, (3) a PET-MAD model which was fine-tuned on the same recomputed water dataset and (4) values for a simple classical force-field that reproduces well most of the experimental properties of water and allows us to demonstrate the slow convergence of some of the estimators[53].

The main takeaways from this comparison (shown in Fig. 7) are that (1) PET-MAD inherits the tendency of GGA functionals to over-estimate the melting point of water, leading to room temperature simulations of a highly undercooled liquid, that is overstructured, with excessive delocalization of protons along H bonds, and a heat capacity that is closer to that of ice than to water; and (2) that the general-purpose model is in excellent agreement with the bespoke models, both in classical simulations and in path integral simulations that are sufficient to converge the pair-correlation function, and approach convergence of the heat capacity. This is significant as it proves that PET-MAD can be used reliably even when probing the large intra-molecular distortions induced by zero-point fluctuations, and that it is fast enough to afford the large overhead of path-integral simulations with complex estimators – even though it cannot escape the limitations of the electronic-structure reference.

## Quantum nuclear effects in NMR crystallography

We now consider the impact of quantum nuclear fluctuations in a different context, evaluating how they affect the nuclear magnetic resonance (NMR) chemical shieldings in organic crystals, and hence how they affect NMR crystallography. In the prototypical NMR crystallography workflow, chemical shieldings are computed for a set of static candidate structures, and compared with those measured experimentally for an unknown polymorph[54]. The most likely candidate structure is then taken to be the one for which the experimentally measured and computed shieldings best agree.

NMR shieldings result from the averaging of instantaneous values of structures distorted by thermal and quantum fluctuations, which can be sampled computationally by performing MLIP-driven (PI)MD simulations and then evaluating the chemical shieldings using a bespoke machine learning model for succinic acid crystals. We

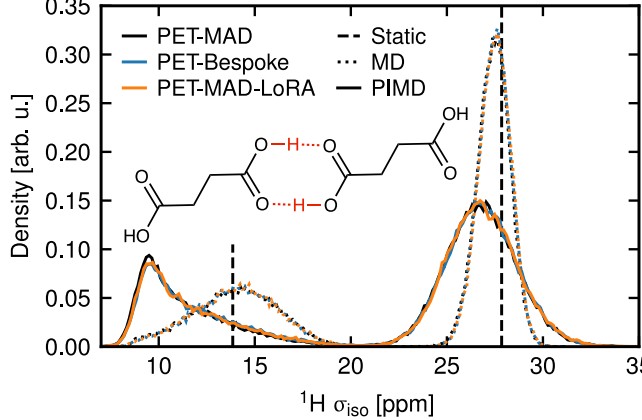

**Fig. 8 | Computed chemical shielding distribution of $^1$H in $\alpha$ succinic acid crystals.** The inset shows the hydrogen bonding pattern in the succinic acid dimer of $\alpha$-succinic acid. The results are obtained using PET-MAD (black), PET-MAD-LoRA (orange) and PET-Bespoke (blue) models for PIMD (solid) and MD (dotted) simulations. Shieldings of the static geometry (PET-MAD only) are shown as vertical lines (carboxylic H3 site around 15 ppm and an average of the aliphatic H1 and H2 sites around 28 ppm).

simulate in particular the $\alpha$ polymorph of succinic acid, one of the structures investigated in a previous study[55] from which we also obtain the training structures to build bespoke shielding and MLIP models (the test set accuracies (MAE) on energy|forces are 12.5 meV/atom|106.1 meV/Å for PET-MAD, 3.1 meV/atom|86.0 meV/Å for PET-Bespoke and 2.0 meV/atom|64.5 meV/Å for PET-MAD-LoRA). In line with the excellent accuracy of PET-MAD for molecular crystals, we observe near-perfect agreement between the distributions of chemical shieldings obtained with the three potentials (Fig. 8), with both classical and quantum sampling – the latter showing a large downward shift and broadening of the $^1H$ shielding distributions, especially for the H-bonded protons, qualitatively the same observation that Engel and coworkers make in ref. 55 using a bespoke MLIP trained on PBE0-MBD reference calculations. As in the case of water, these results demonstrate the ability of PET-MAD to describe quantum nuclear fluctuations. By combining MLIP-driven sampling with models of functional properties, such as the chemical shieldings, one can extend the reach of a general-purpose model beyond structural and energetic predictions.

## Dielectric response of barium titanate

To conclude our benchmarking series, we study the ability of PET-MAD to describe the dielectric properties of materials using barium titanate system as an example. Barium titanate (BaTiO$_3$, BTO) is a prototypical ferroelectric perovskite that undergoes a sequence of temperature-dependent structural phase transitions: rhombohedral (R, below 183 K), orthorhombic (O, 183–278 K), tetragonal (T, 278–393 K), and cubic (C, above 393 K)[56]. The ferroelectricity in BTO arises from off-center displacements of the Ti atom within the oxygen octahedron, which break centrosymmetry in the lower-temperature phases. These displacements are fundamental to its ferroelectric behavior and phase transitions[57], making BTO an ideal model system for studying ferroelectricity in perovskites.

To evaluate the capability of PET-MAD in characterizing ferroelectricity in this material, we perform flexible-cell MD simulations for a 320-atom model of BTO over a temperature range of 40–400 K at constant ambient pressure, following the same protocol as in ref. 58 and as reported in the Methods. Once again, we recompute consistent energetics for the material-specific dataset, to build bespoke and fine-tuned PET models. Test-set MAEs on energy|forces are 12.67 meV/atom|27.96 meV/Å for PET-MAD, 0.23 meV/atom|9.41 meV/Å for the bespoke model, and 0.12 meV/atom|3.92 meV/Å for the LoRA-fine-tuned model, respectively.

We then analyze the sampled configuration within a reduced-dimensionality collective variable landscape derived from a subset of neighbor density coefficients[58] (shown in the inset of Fig. 9). Measuring the populations of different phases by a clustering algorithm allows us to estimate their relative chemical potential, and hence the transition temperatures between the different phases. Similar to what is observed in ref. 58, the transition temperatures are heavily underestimated with respect to experiments (due to a combination of DFT shortcomings and finite-size effects), but there is excellent agreement between PET-MAD and the two bespoke models. The largest discrepancy, for the T-C transition, is below 30 K−a remarkable accuracy for a model with such broad applicability.

To further assess the reliability of PET-MAD in predicting functional properties, we also compute the temperature-dependent static dielectric tensor $\epsilon_r$ (Fig. 9), by estimating the covariance of the total polarization, computed in turn using an equivariant linear model trained on the polarization dataset of ref. 58 (more details in the Methods). All models predict qualitatively the large value of $\epsilon_r$ for the paraelectric, cubic phase, which increases greatly as it approaches the ferroelectric transition temperature, and then takes smaller values in the tetragonal and orthorhombic phases (which have multiple inequivalent optical axes). As it is the case of the transition

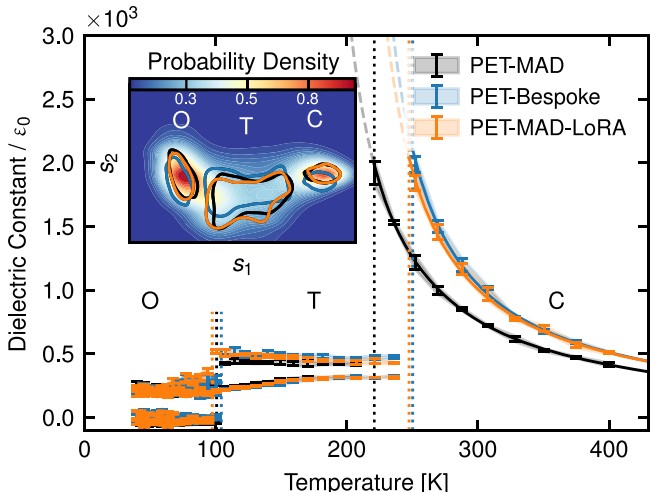

**Fig. 9 | Temperature-dependent dielectric response of BTO.** The results for three PET models−the pre-trained PET-MAD, the bespoke PET model, and a LoRA-fine-tuned PET-MAD−are shown in black, blue, and orange lines, respectively. Vertical dotted lines correspond to phase transition temperatures estimated from the relative chemical potentials. Error bars correspond to standard deviations across four independent MD runs. The inset shows the collective variable landscape distinguishing three separate phases, with the corresponding probabilities of occupying each phase. The models consistently identify the different phases, as evidenced by the contour plots at 0.5 probability centered on the basins corresponding to each phase.

temperatures, the two specialized models are in near-perfect agreement, while the PET-MAD simulations slightly underestimate $\epsilon_r$ in the T phase, and yield a curve in the C phase that is shifted by about 25 K, consistent with the lower Curie temperature. Overall, this final example demonstrates once more that, despite considerably larger test errors than for a bespoke model, PET-MAD captures with close-to-quantitative accuracy subtle physical effects, and can be used to drive complicated, advanced materials modeling protocols.

## Discussion

PET-MAD pushes the boundaries of what generally-applicable universal machine learning potentials can achieve, providing out-of-the-box semi-quantitative accuracy for both inorganic and organic materials benchmarks, as well as for six demonstrative applications covering a very broad range of material types and advanced simulation techniques. Moreover, it does so while being trained on a data set that is orders of magnitude smaller than those used for current state-of-the-art models. This observation suggests that the computational budget for training set construction can be best allocated by pursuing internal consistency and a high degree of structural and chemical diversity − two guiding principles which underlie the construction of the MAD dataset we introduce here. A compact dataset facilitates model training and optimization, as well as the transition to more accurate and computationally demanding approximations to the electronic-structure problem.

The PET architecture we use is not constrained to follow rigorous rotational invariance; nonetheless, it learns to make predictions that are invariant to a high degree of accuracy. It achieves a high expressivity thanks to its transformer module, while being faster than all the conservative models we considered, and even faster when using non-conservative forces. The availability of a reliable and inexpensive uncertainty quantification framework makes it possible to assess model error and also its propagation to the ultimate quantity of interest. In cases where this error is larger than desired, a simple fine-tuning procedure can be applied to further improve the accuracy for a

specific target system, while maintaining the ability to make qualitative predictions for the high-diversity test set. PET-MAD is easily accessible from several atomic-scale simulators thanks to its integration within the `metatensor`[59] ecosystem. It provides an efficient and accurate solution to perform simple and advanced atomistic simulations of materials, and we expect it will serve as inspiration for future developments of both datasets and model architectures, as well as a reliable engine to drive the design and discovery of new and better materials.

## Methods

We provide a brief but complete overview of the procedure we followed to construct the MAD dataset, to train the PET-MAD model, and to fine-tune single-purpose models. The simulation protocols for the advanced simulation examples follow closely those of the cited publications, and are briefly discussed in the SI for the convenience of the reader.

### MAD dataset construction

In four of the MAD subsets, namely MC3D[23], MC2D[24], SHIFTML-molcrys[25], and SHIFTML-molfrags[27], we simply used the published structures, recalculating their energy and forces with the MAD DFT settings provided in "First-principles calculations". For the remaining MC3D-based subsets, we always started from the original MC3D crystals and made different structural modifications, with the sole goal of improving coverage of the configuration space for highly-distorted, unusual configurations. Structures in the MC3D-rattled subset were obtained by selecting each MC3D crystal with more than one atom in the unit cell and applying to the Cartesian coordinates of each atom a Gaussian noise with zero mean and a standard deviation equal to 20% of the corresponding covalent radii. The MC3D-randcomp subset was created by first taking the atomic positions and cell parameters of a random crystal structure from the MC3D subset, then assigning random atom types to every lattice site, and finally adjusting isotropically the total volume of the cell to match the total atomic volume calculated based on the covalent radii of the elements. The surface slabs in the MC3D-surface subset were obtained by cleaving a randomly-selected MC3D crystal along a random, symmetrically distinct crystallographic plane with the maximum value of the Miller index (hkl) equal to 3, also ensuring the orthogonality of the normal lattice vector to the surface plane. The MC3D-cluster subset was constructed by cutting a random spherical atomic environment containing between 2 and 8 atoms from a randomly selected MC3D crystal structure.

### First-principles calculations

All datasets in this work (i.e., MAD and the reference subsets used for benchmarking, as well as the single-purpose dataset for the six case studies) were built by computing energies and forces using an identical density functional theory (DFT) setup. We used the Quantum Espresso v7.2 package[60], compiled with the SIRIUS v7.5.2[61] libraries as the primary DFT engine and AiiDA v2.6.3[62] as a workflow and task manager. To reduce the possibility that some simulations would converge to ambiguous magnetization states, all simulations were performed in a non-magnetic setting, using the PBEsol exchange-correlation functional. Even though this choice is known to have shortcomings for several classes of materials – as shown by the discrepancy of some of our examples with experiments – it allows us to generate predictions for a highly diverse system with a high degree of stability and converged numerical parameters. We described valence and semi-core electrons using the standard solid-state pseudopotentials library (SSSP) v1.2 (efficiency set)[63], with the most stringent settings for plane-wave and charge-density cutoffs across the 85 elements (110 Ry and 1320 Ry, respectively). As discussed in the Supplementary Information, using different settings to match the maximum cutoff of each structure would give substantial inconsistencies between the description of some atomic types. Electronic smearing and partial occupancies were

described with a Marzari-Vanderbilt-DeVita-Payne cold smearing with a spread of 0.01 Ry. In all periodic dimensions, the first Brillouin zone was sampled with a Γ-centered grid with a resolution of 0.125 Å⁻¹, while only the Γ point was used along the non-periodic dimensions. In the case of non-periodic systems, we also applied a truncation scheme of the Coulomb potential to avoid the interaction of periodic replicas of the system through the periodic boundary conditions: the Sohier-Calandra-Mauri method for 2D systems and the Martyna-Tuckerman correction for 0D systems. Along each non-periodic direction, we additionally included a 25 Å vacuum to ensure convergence of the truncation methods. A compositional baseline based on isolated atom energies was subtracted from the DFT energies of each structure to improve the numerical stability of training.

For most of the MAD subsets listed in the MAD dataset, the DFT settings provided above ensured a good convergence rate of > 95%, i.e. convergence was not achieved in less than 5% of the cases. The exception is MC3D-random, for which only about 55% of the attempted simulations converged. This result does not come as a surprise and can be explained by the highly distorted state of these structures.

### PET model

As our machine learning approach for fitting interatomic potentials, we employ the Point Edge Transformer (PET)[18], which has been shown to achieve state-of-the-art performance across a diverse set of benchmarks spanning both molecular and materials domains. In essence, PET is a graph neural network (GNN) where each message-passing layer is implemented as an arbitrarily deep transformer. Specifically, PET maintains feature vectors (or messages) $f_{ij}^l$ for every directed bond between atoms $i$ and $j$ that lie within a specified cutoff radius. Here, the superscript $l$ denotes the message-passing layer index, and each $f_{ij}^l$ is a fixed-size vector (a token) of dimension $d_{PET}$. These intermediate representations are updated at each message-passing layer by a transformer with an adjustable number of internal layers, invoked for each atomic environment. At each message-passing layer, for each atom $i$, the input tokens to the transformer consist of the message vectors $\{f_{ji}^l\}_j$ coming from all neighbors $j$ to the central atom $i$. The transformer then applies a sequence-to-sequence transformation in a permutation-covariant manner. Its outputs are subsequently interpreted as the new set of outbound messages from atom $i$ to each neighbor $j$, $\{f_{ij}^{l+1}\}_j$. Geometric information (i.e., the 3D positions of neighbors) and chemical species are also incorporated into this process. To obtain the desired target property (e.g., energy), PET applies feed-forward neural networks to each representation $f_{ij}^l$ and sums all the outputs across bonds $ij$ and layers $l$. The PET architecture imposes no explicit rotational symmetry constraints, but learns to be equivariant through data augmentation. This unconstrained approach yields high theoretical expressivity: even a single layer of the model acts as a universal approximator featuring virtually unlimited body order and angular resolution. A detailed specification of the architecture and the full functional form can be found in ref. 18.

### Training of PET-MAD

We split the MAD dataset into training, validation, and test parts by separately shuffling each of the eight subsets, and selecting respectively 80%, 10%, and 10% of the structures. The composition contribution to the total energy of the structures was fitted using a simple linear model and subtracted to improve the training behavior. We used a Pareto-optimal architecture (see the details on hyperparameter optimization in the Supplementary Information) with a cutoff radius of 4.5 Å, 2 message-passing layers, each containing 2 transformer layers with a token size of 256, 8 heads in the multi-head attention layer, 512 neurons in the output multi-layer perceptron. Training was performed using the PyTorch framework and the `metatrain` package[59] on 8 NVIDIA H100 GPUs with a batch size of 24 structures per GPU for a total of 1500 epochs and took about 40 hours. We used the Adam optimizer

with an initial learning rate (LR) of $10^{-4}$ and applied an LR scheduler, which halved the LR every 250 epochs. The loss function was based on a root-mean-square error difference between the model's energy and force predictions and the corresponding target values, with a scaling factor of 0.1 applied to the energy contribution. Mean absolute errors of the trained PET-MAD model were 7.3 meV/atom|43.2 meV/Å in energies and forces prediction on the train set. Validation set errors were 14.7 meV/atom|72.2 meV/Å, respectively.

## Fine-tuning

Fine-tuning of the pre-trained PET-MAD model was performed using a parameter-efficient fine-tuning technique based on low-rank adaptation (LoRA)[28]. This method is widely used in the ML community, since it allows to mitigate *catastrophic forgetting* − the phenomenon of losing accuracy on the base dataset while tuning the model on the new dataset, which is inherent to conventional fine-tuning, where all the weights of the model are trainable[64]. In the LoRA approach, the weights of the base model are frozen during the training, while an additional set of trainable weights composed of two low-rank matrices is added to each attention block of the model with a regularization factor, which controls the influence of the low-rank matrices on the model's weights. By adjusting rank and regularization factor, one can optimize the fine-tuning workflow to achieve better performance on specific tasks, without compromising entirely the ability to perform the more general task. In this work, we used a rank of 8 and a scaling parameter of 0.5 for all LoRA fine-tuned models, unless otherwise specified.

In terms of raw accuracy on the specialized datasets, the evidence in "Results" and Supplementary Note 11 indicate that fine-tuning is always beneficial compared to training a specialized ad hoc model in the low-data regime. Even though on larger datasets a specialized model trained from scratch can exceed the accuracy of a fine-tuned model, this is not the case for many of the systems investigated in this work, including barium titanate, succinic acid, lithium thiophosphate, and high-entropy alloys.

LoRA-fine-tuned models retain a varying degree of accuracy (see the Supplementary Note 10 for details) on the generic structures from the MAD dataset, while still providing practically equivalent observables from an ad-hoc fully specialized model. In general, we therefore recommend employing LoRA if fine-tuning PET-MAD for a specific application.

## Uncertainty quantification

We provide uncertainty quantification for the PET-MAD model via the LLPR method[29], which produces a posteriori uncertainty estimates for trained neural networks. The LLPR uncertainties can be computed based on the covariance of the last-layer features over the training set, and they can be evaluated at nearly no additional cost compared to the raw predictions. The LLPR formalism is particularly suitable in the context of this study, as it can be used to sample a finite number of last-layer weight sets. The resulting last-layer ensemble can be used to propagate uncertainties through arbitrarily complex atomistic workflows, by manipulating separately the outputs of the different ensemble members, and/or using them to reweight the trajectories generated by the mean of the ensemble[44].

## Simulations details

In this section, we provide an overview of the methodology to explain the technical challenges and provide a few details that can be useful to appreciate the implications of the benchmark accuracy. Unless otherwise specified, the simulations were performed using the LAMMPS package (27 Jun 2024) with Kokkos version 4.3.1[65].

**Ionic transport in lithium thiophosphate.** The ionic conductivities, $\sigma$, were computed via the Green-Kubo theory of linear response[41], which is a practical framework to compute transport coefficients of extended

systems[39]. For an isotropic system of $N$ interacting particles:

$$\sigma = \frac{\Omega}{3k_{\mathrm{B}}T} \int_0^\infty \left\langle \mathbf{J}_q(\Gamma_t) \cdot \mathbf{J}_q(\Gamma_0) \right\rangle dt, \tag{1}$$

where $k_{\mathrm{B}}$ is the Boltzmann constant, $T$ the temperature and $\Gamma_t$ indicates the time evolution of a point in phase space from the initial condition $\Gamma_0$, over which the average $\langle \cdot \rangle$ is performed. $\mathbf{J}_q$ is the charge flux, which depends only on the velocities of the atoms, $\mathbf{v}_i$, and their charges, $q_i$:

$$\mathbf{J}_q = \frac{e}{\Omega} \sum_i q_i \mathbf{v}_i. \tag{2}$$

Here, the sum runs over all the atoms, $e$ is the electron charge, and the $q_i$ are equal to the nominal oxidation number of the atoms.

In order to simulate the charge transport in LPS, in Fig. 4, we perform MD simulations in the NPT ensemble of a quasi-cubic 768-atom cell in all stable $\alpha$, $\beta$, and $\gamma$ phases with a constant isotropic pressure of $p = 0$ atm for a set of temperatures between 450 K and 900 K. The system's center of mass is kept fixed during the simulations, and the timestep was set to 1 fs. For each simulation, we first run 200 ps of thermalization, then we start collecting the ionic current for 4 ns.

**Melting point of GaAs.** The melting points were computed from interface pinning simulations of GaAs liquid-solid interface model structures, following Imbalzano's work[43]. At the melting point, the chemical potential of the liquid phase $\mu_l$ and the solid phase $\mu_g$ are identical. Applying a pinning potential restrains the system to coexisting liquid and solid phases separated by a planar interface. A suitable collective variable must be chosen that distinguishes between liquid and solid phases - we employed the same atom-centered Steinhardt Q4 order parameter as chosen by Imbalzano. For a given configuration $A$, the expression for the pinning potential $V_s(A)$, reads:

$$V_s(A) = k/2(s(A) - \bar{s})^2 \tag{3}$$

where, $k$ is the spring constant of the restraint, $s(A)$ is the value of the collective variable for a given configuration (the sum over all atomwise Q4s), and $\bar{s}$ is the value of the collective variable to which the configuration is restrained. When $s(A)$ and $\bar{s}$ are normalized with respect to fully solid and liquid boxes, setting $\bar{s}$ to $\frac{1}{2}$ restrains the system to a half-liquid half-solid box. The average force of the pinning potential acting on the system is proportional to the chemical potential difference $\Delta\mu$ between the pinned liquid and solid phase at a given temperature:

$$\Delta\mu \propto k \left\langle s(A) - \bar{s} \right\rangle \tag{4}$$

The melting point is then determined via root finding by simulating pinned systems at increasing temperatures and determining when the average force is zero. We compute the collective variable with the PLUMED package and run molecular dynamics with the LAMMPS package to determine the average pinning force. Interface structures are constructed by first building elongated supercells (1152 Atoms, 17 Å × 17 Å × 90 Å) of GaAs and partial melting of half of the box, whilst constraining the solid half. These simulations were used to determine the average restraining value $\bar{s}$. The supercells were relaxed with the respective PET potential and then partially melted with NPT MD runs. Production MDs were run for 1 ns, using a 4 fs timestep. The volume is kept fixed in the x and y directions while a barostat is applied on the z direction to fix the pressure at the interface around 1 bar. Simulations were performed for temperatures from 950 K to 1200 K in intervals of 50 K.

We propagate the model uncertainties of the potential energies to the uncertainties of the melting point via thermodynamic reweighting

of the observables computed from the interface pinning simulations that were driven by the respective PET potential. Through the LLPR ensemble uncertainty quantification scheme, we obtain at each time-step of the simulation a committee of $N$ potential energy predictions (where $N = 128$ is the number of sampled LLPR ensembles), which are then used to reweight the instantaneous value of the collective variable. The reweighted collective variables are then averaged across the trajectory for each committee member, yielding $N$ chemical potential differences. From the chemical potential differences, we then compute a set of $N$ melting points, which we either use as a nonparametric estimate of the distribution of the computed melting point or take the standard deviation of the reweighted melting points as an uncertainty estimate of the computed melting point. A more detailed description of using thermodynamic reweighting for uncertainty quantification of thermodynamic averages can be found in ref. 44.

**Surface segregation in high-entropy alloys.** To study the surface segregation in the CoCrFeMnNi alloy, we prepared a surface slab with a *fcc* lattice in the (111) orientation and a $7 \times 7 \times 11$ supercell containing 539 atoms. Relaxation of both structure and composition of the surface was performed within a replica-exchange molecular dynamics run with Monte-Carlo atom swaps with 16 replicas for 200 ps in the NPT ensemble using a 2 fs timestep at zero pressure and a logarithmic temperature grid ranging from 500 K to 1200 K.

We used both the pre-trained PET-MAD and fine-tuned PET-MAD potentials to perform identical REMD/MC runs. The fine-tuning step was performed starting from PET-MAD model weights and training on a subset of the HEA25S dataset[48] containing 2000 randomly chosen structures (1000, 500, 200, 200, and 100 entries from the subsets O, A, B, C, and D of the HEA25S dataset, respectively), recalculated using MAD DFT settings (see MAD dataset). Details of fine-tuning are provided in the Supplementary Information.

The surface composition of the (111) surface of the relaxed CoCrFeMnNi alloy was analyzed if terms of Gibbs surface excess per unit area $\Gamma_a$, which indicates the surface segregation propensity of the elements in the alloy. It is defined as

$$\Gamma_a = \frac{N_a - N_a^B \cdot N/N^B}{S}, \tag{5}$$

where $N_a$ and $N_a^B$ correspond to the total number of atoms of element $a$ in the slab, and the number of atoms of element $a$ inside the bulk region of the surface slab. $N$ and $N^B$ represent the total numbers of atoms in the cell and inside the bulk region, respectively, and $S$ is the surface area. The bulk region is defined as a 10Å-thick region around the center of the slab. Therefore, the values of $\Gamma_a > 0$ correspond to enrichment of the surface with element $a$ compared to bulk of the material, while $\Gamma_a < 0$ in contrast, correspond to surface depletion. If $\Gamma_a \approx 0$, the concentration of element $a$ at the surface and in the bulk is roughly the same. Defined in this way, $\Gamma_a$ allows the surface composition to be analyzed independently of the choice of bulk layer thickness, providing a macroscopic measure of surface affinity.

**Quantum nuclear effects in liquid water.** The most widespread method to include nuclear quantum effects for equilibrium observables at a constant temperature $T$ is path integral molecular dynamics[50] (PIMD). In this variant of molecular dynamics, a number $P$ of equivalent replicas of the system are run simultaneously at temperature $PT$. The system is evolved classically, according to an overall potential which is the sum of the potential energies of the individual replicas, plus a harmonic spring term between all corresponding atoms belonging to adjacent replicas:

$$V'\left(\{\mathbf{r}\}_{j=1}^P\right) = \sum_{j=1}^P V(\mathbf{r}_j) + \frac{1}{2}\omega_P^2 \left|\tilde{\mathbf{r}}_j - \tilde{\mathbf{r}}_{j-1}\right|^2, \tag{6}$$

where $V'(\{\mathbf{r}\}_{j=1}^P)$ is the potential energy of the extended classical system, $j$ indexes the $P$ replicas (also called beads) of the system, and $V(\mathbf{r})$ is the potential energy surface. Furthermore, $\tilde{\mathbf{r}}$ contains the mass-scaled positions ($\tilde{r}_i = r_i\sqrt{m_i}$ for every atom $i$) and we define $\tilde{\mathbf{r}}_0$ so that it "wraps around" to $\tilde{\mathbf{r}}_P$. The angular frequency of the harmonic terms between beads is given by $\omega_P = k_BPT/\hbar$.

The treatment of NQEs for equilibrium observables calculated with PIMD is exact as $P \rightarrow \infty$, and it reduces to that given by classical MD for $P = 1$. The calculation of thermodynamic averages from PIMD simulations involves the use of so-called "estimators"[50], which can range from simple averages over the bead positions for position-dependent observables to more complicated expressions for momentum-dependent observables. In this work, we use the heat capacity estimators from ref. 51.

All simulations were run with a box of 128 water molecules at constant volume and temperature. The density was chosen to be 997.1 kg/m³ (corresponding to the experimental density of water at 298 K and 1 atm), and the temperature was set to 298 K. i-PI[66] (version 3.1) was used with LAMMPS as its backend in order to enable PIMD and evaluate path integral estimators.

**Quantum nuclear effects in NMR crystallography.** We compute thermodynamic averages of chemical shieldings from MD and PIMD simulations following the exact computational protocol described in ref. 55, replacing bespoke MLIPs with the PET MAD potentials (Bespoke, PET-MAD-LoRA, PET-MAD). MD and PIMD simulations were performed with a timestep of 0.5 fs and a total simulation duration of 250 ps in the NPT ensemble, applying an external isotropic pressure of 1 bar at 300 K. PIMD simulations were run with 32 beads. We obtain thermodynamic averages of chemical shieldings from the trajectories using bespoke surrogate models that we train on an existing dataset of GIPAW shielding reference calculations of succinic acid crystals from ref. 55. Exact convergence parameters can be taken from ref. 55 and can be regarded as converged for the chemical shielding computations of organic solids. We fit one linear model considering the local atomic environments of all hydrogen atoms in succinic acid using SOAP descriptors and obtain models of similar accuracy as described in ref. 55. The $^1$H $\sigma_{iso}$ test set errors (RMSE) is 0.17 ppm for our linear model compared to 0.16 ppm of the kernel model from the original publication[55]. A parity plot of the shielding predictions against the GIPAW reference values can be found in Supplementary Fig. 14.

**Dielectric response in barium titanate.** Pressure and temperature along MD simulations are controlled through the Nosé-Hoover barostat and stochastic velocity rescaling, respectively. The system size of 320 atoms corresponds to a $4 \times 4 \times 4$ supercell of the 5-atom BTO unit cell. The different structural phases are identified by clustering a reduced-dimensionality representation of the sampled structures with a Gaussian mixture model, which assigns to every sampled structure, $t$, a set of probabilities, $P_k(t)$, corresponding to each identified cluster $k$, effectively labeling the phases[58]. The phase transition temperature is estimated by evaluating the relative chemical potential between two phases, $k$ and $k'$:

$$\Delta\mu^{kk'}(T) = -k_B T \log\frac{\sum_t P_k(t)}{\sum_t P_{k'}(t)}. \tag{7}$$

At phase coexistence, $\Delta\mu^{kk'} = 0$, yielding a practical way to determine the phase transition temperature. In practice, we perform a linear fit of $\Delta\mu^{kk'}$ as a function of $T$ to locate the temperature where $\Delta\mu^{kk'}(T) = 0$.

The relative static dielectric tensor is computed from the covariance of the cell dipole moment, **M,**

$$\varepsilon_{r,\alpha\beta} = \delta_{\alpha\beta} + \frac{\text{cov}(M_\alpha, M_\beta)}{\varepsilon_0 \Omega k_B T}, \quad (8)$$

where $\varepsilon_0$ is the vacuum permittivity, $\Omega$ is the system volume, $k_B$ is the Boltzmann constant, $\alpha$, $\beta$ denote Cartesian directions, and $\delta_{\alpha\beta}$ is the Kronecker delta. In the cubic phase, as identified by the clustering algorithm, the average dipole is assumed to be zero due to centrosymmetry. In this phase, the dielectric tensor is proportional to the identity matrix, effectively reducing to a scalar quantity. As the system transitions from the cubic to the tetragonal phase, the dielectric components parallel and perpendicular to the polarization axis become anisotropic, with dipole fluctuations suppressed along the polarization axis. Upon further cooling and the transition from the tetragonal to orthorhombic phase, all remaining symmetries are broken (except for those under permutation of Cartesian directions), resulting in six independent components.

The cell dipole moments are obtained by evaluating an equivariant $\lambda$-SOAP-based linear model trained on the dipole dataset from ref. 58. These dipoles are then used to compute the dielectric tensors using Eq. (8). The parallel and perpendicular components, used for plotting the dielectric tensor in the tetragonal phase, are computed by projecting the dielectric tensor onto the dipole vector and its orthogonal complement, respectively:

$$\varepsilon_\parallel = \sum_{\alpha\beta} \varepsilon_{r,\alpha\beta} \frac{M_\alpha M_\beta}{|\mathbf{M}|^2}, \quad (9)$$

$$\varepsilon_\perp = \sum_{\alpha\beta} \varepsilon_{r,\alpha\beta} \left( \delta_{\alpha\beta} - \frac{M_\alpha M_\beta}{|\mathbf{M}|^2} \right). \quad (10)$$

## Data availability
All data generated in this study have been deposited in the Materials Cloud Archive database under accession code materialscloud:2025.145 at https://doi.org/10.24435/materialscloud:fe-1p.

## Code availability
All code contributions are available as public *git* repositories, with simplified installation instructions available at https://github.com/lab-cosmo/pet-mad. The standalone version of the code is available at https://doi.org/10.5281/zenodo.17171506. A practical usage example can be found at http://atomistic-cookbook.org/examples/pet-mad/pet-mad.html.

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

## Acknowledgements

The Authors would like to thank Giovanni Pizzi, Marnik Bercx and Sebastian Huber for helping with the set-up of AiiDA calculations on the MC3D dataset, and Sandip De for discussion on potentials for high-entropy alloys. They are also grateful to all the current and past members of the Laboratory of Computational Science and Modeling who contributed to the software infrastructure that supported this work. Computation for this work relied on resources from the Swiss National Supercomputing Centre (CSCS) under the projects s1092, s1219, s1243, s1287, lp26, Finnish IT Center for Science (CSC) under the project 465000551, and the EPFL HPC platform (SCITAS). AM and MC acknowledge support from an Industrial Grant from BASF and from EPFL. GF and MC acknowledge funding from the MARVEL National Centre of Competence in Research (NCCR), funded by the Swiss National Science Foundation (SNSF, grant number 182892) SP, FB and GF Platform were supported by a project within the Platform for Advanced Scientific Computing (PASC). DT acknowledges support from a Sinergia grant of the Swiss National Science Foundation (grant ID CRSII5_202296). PL, PP and MC acknowledge the funding from the

European Research Council (ERC) under the European Union's Horizon 2020 research and innovation program (grant agreement No 101001890-FIAMMA). MK and MC acknowledge support by a Swiss National Science Foundation (grant ID 200020_214879).

## Author contributions

A.M. worked on the creation of the MAD dataset, training the PET-MAD model, running the accuracy benchmarks for PET-MAD and reference models, and performing the calculations of surface segregation in CoCrFeMnNi. F.B. worked on the implementation of the `metatrain` infrastructure for training and evaluating the PET-MAD model, as well as performing the uncertainty quantification for it, performing the speed benchmarking for PET-MAD and reference models, and studying the nuclear quantum effects in water. M.K. calculated the melting point in GaAs and studied the chemical shielding in succinic acid. P.P. studied the dielectric response in barium titanate and benchmarked geometry optimization and phonon predictions. D.T. studied the ionic conductivity in lithium thiophosphate. G.F. developed the `metatensor` ecosystem, which `metatrain` infrastructure is based on, developed the interfaces for ASE and LAMMPS simulation engines, and helped with the computational setup for creating the MAD dataset. S.P. developed the original version of the PET architecture. P.L. worked on the implementation of the `metatrain` infrastructure and finalized the figures. M.C. designed and guided the project and provided theoretical support. All authors contributed to the original version of the manuscript.

## Competing interests

The authors declare no competing interests.
