## [Transparent Peer Review file · Nature Communications]

PET-MAD as a lightweight universal interatomic potential for advanced materials modeling

Corresponding Author: Professor Michele Ceriotti

Version 0:

Reviewer comments:

Reviewer #1

(Remarks to the Author)

Dr. Michele and coauthors present a new forcefield together with a training set which covers materials other than bulk systems. The work is solid. However, I did not find the work presenting any significantly novel ideas, neither does it show significant performance gain compared with SOTA. Therefore, I do not find it, in the current form at least, acceptable for Nat Comm but more like a fit to more specialized journals. However, I would still suggest giving it a shot - let authors rework on the organization of the paper to better present its novelty. I'm happy to see the modified version for any new insights. In addition to this, there are a few aspects that concern me quite a lot, see below:

1. Some claims are not true even in the abstract, say, "recent "universal" models deliver qualitative accuracy across the periodic table but are often biased toward low-energy configurations." This was true for early models like M3GNet and MACE-MP-0. Recent open datasets already cover non-local minima structure quite extensively, like META's OMat24. The related models perform really well under high-temperature regimes.
2. "PET-MAD rivals state-of-the-art MLIPs for inorganic solids, while also being reliable for molecules, organic materials, and surfaces." This is an overly generalized big claim. I agree that for some properties, PET-MAD might have achieved SOTA but definitely not for all of the metrics. Just to put here the test of the performance on MatBench Discovery's thermal conductivity. PET-MAD is certainly not the best (the authors can retest this for more reliable results)... I'm not saying that this benchmark is perfect or should it represent the overall performance of a universal forcefield. In any case, it is certainly one most popular benchmark. The current result clearly is a counterexample to "PET-MAD rivals state-of-the-art MLIPs for inorganic solids" The authors need to be specific on such big claims, especially on the performance side...

Model ksrme
eSEN-30M-OAM 0.17
ORB v3 0.21
SevenNet-MF-ompa 0.317
GRACE-2L-OAM 0.294
eSEN-30M-MP 0.34
MACE-MPA-0 0.412
MatterSim v1 5M 0.574
DPA3-v2-OpenLAM 0.687
GRACE-1L-OAM 0.516
SevenNet-l3i5 0.55
MatRIS v0.5.0 MPtrj 0.861
GRACE-2L-MPtrj 0.525
DPA3-v2-MPtrj 0.959
MACE-MP-0 0.647
PET-MAD 1.244
AlphaNet-MPTrj 1.31
eqV2 M 1.771
ORB v2 1.732
eqV2 S DeNS 1.676
ORB v2 MPtrj 1.725

3. Apart from smaller size, how is the dataset design principle different from the one from PFP? Towards universal neural network potential for material discovery applicable to arbitrary combination of 45 elements | Nature Communications This needs to be articulated better as the authors have already done surfaces, Molecules, bulk, disordered systems et al. and on a large scale.....
4. Figure 1 is not very easy to read... the colors are too close. The presentation needs to be changed. Also, it is not fair to compare in-distribution with other models' error where the test set is randomly chosen from the same pool of training set. The authors either should use some open benchmark or need to create a new benchmark that is independent of the training set distribution, aka, using a different rule to construct it. Examples could be: a. some bulk from Alex; b. some molecular data from MD17; 3. Some surface data from OC22; 4. Some off-local-minima data from MatterSim's codebase et al. Things like these reflect the performance better.
5. In Fig.2, authors can safely remove non-conservative models like Orb. No need to compare until they get to the same level of performance to conservative models. Or use another way to present non-conserve models. The current presentation is a bit undercovering the contribution from this work as PET-MAD seems quite fast.
6. PAGE 9 is very concerning. Again, the benchmarks should NEVER come from the construction of a training set, even if it is randomly split out. It is not a fair benchmark. Therefore, I highly doubt the current claims in these entire sections. New benchmarks need to be set up in statistically irrelevant manner compared with the training set.
7. The dataset is constructed using PBEsol. Is there any particular reason for that? I initially thought this would be something like metaGGA or hybrid when I read the abstract. But it is not. The novelty needs to be clearly stated here then.... One point might be this is the first opensource dataset with molecules, surfaces, bulk et al with same level of theory, which is plausible. But this certainly is not enough for Nat Comms...
8. Consider giving a distribution of the elements of the training set. Some description on the number of atoms in the training set, the energy and force distribution will let readers get a better overall picture of the dataset.

The manuscript "PET-MAD, a universal interatomic potential for advanced materials modeling" by Dr. Michele and coauthors presents PET-MAD, a machine learning interatomic potential (MLIP) trained on the newly introduced MAD dataset, which includes a wide range of materials beyond bulk crystalline solids, such as surfaces, nanoclusters, molecular crystals. The authors claim that PET-MAD rivals the performance of state-of-the-art MLIPs while providing broader material coverage, and they showcase several application examples.

While the MAD dataset and the PET-MAD model may offer useful resources for the community, after careful evaluation, we do not believe this manuscript meets the standards of novelty required for publication in Nature Communications. Below, I outline the major reasons for my recommendation to reject the manuscript.

1. Overstatement of Claims and Lack of Rigorous Benchmarking

Several claims made in the manuscript, including the abstract, are overstated or insufficiently supported. For example, the statement that "recent 'universal' models deliver only qualitative accuracy across the periodic table but are biased toward low-energy configurations" is no longer accurate for the latest generation of datasets and models (e.g., META's OMat24). No testing against community-accepted, rigorous benchmarks was performed, such as Matbench-Discovery (<https://matbench-discovery.materialsproject.org/>), PhononDB (see <https://github.com/atztogo/phonondb?tab=readme-ov-file>) and the phonon benchmark paper: <https://arxiv.org/abs/2412.16551>), MLIP-Arena (<https://huggingface.co/spaces/atomind/mlip-arena>), or thermal conductivity datasets (developed in <https://arxiv.org/abs/2408.00755> and maintained as part of matbench-discovery). Instead, the benchmarks rely on splits from the training distribution, which does not provide a fair or meaningful assessment of the model's generalization capabilities. Without benchmarking on independent test sets and tasks, the manuscript's performance claims are not credible.

2. Insufficient Novelty in Dataset Construction

Although the MAD dataset extends across molecules, surfaces, and bulk systems, it is unclear how it meaningfully differs from or advances beyond existing datasets, such as those accompanying the "Towards universal neural network potential for material discovery" work published in Nat. Comm. Apart from the reduced size, the design principles and novelty compared to existing efforts are not sufficiently articulated.

Moreover, the choice of PBEsol as the computational level of theory is neither well-motivated nor clearly justified.

Historically, the community has primarily relied on the PBE functional to generate large-scale datasets (e.g. Materials Project, Google's Genome, Microsoft's MatterSim, Meta's OMat), comprising hundreds of millions of data points, with a recent trend to shift to more accurate functionals such as r^2 SCAN (<https://arxiv.org/abs/2503.04070>). The authors should provide a clear rationale for selecting PBEsol.

3. Technical Issues in the Results and Analysis

There are several inconsistencies and technical issues in the results presentation:

- The behavior of the fine-tuned PET-MAD model for the gamma phase at low temperatures (below 625 K) diverges unexpectedly from both the bespoke model and the pretrained PET-MAD model. This inconsistency is unexplained and raises concerns about the model's fine-tuning robustness. Typically, the fine-tuned model should give similar or the same results as the bespoke model, but this is not the case in this figure.
- In the water simulation example, no comparison is made to experimental data or first-principles MD simulations (e.g., using PBE). Without these comparisons, it is difficult to assess the physical reliability of the model for molecular systems. Water systems have been extensively studied, and experimental radial distribution functions and angle distributions (e.g., O—O—O angles) are well-known references. How closely does the fine-tuned PET-MAD model reproduce these established results?
- Fine-tuning the PET-MAD model to a water dataset with the same level of theory is a relatively easy task. A truly universal model should be able to fine-tune toward higher-level datasets (e.g., revPBE0-D3), which would better demonstrate the expressiveness of the pretrained representations. For example, the authors may refer to the recent MatterTune manuscript (<https://arxiv.org/abs/2504.10655>).

4. Presentation Issues

Several figures require major improvements for clarity and fairness:

- Figure 1 uses colors that are too similar and visually confusing. More importantly, comparing in-distribution test errors against other models trained and tested independently is misleading. Proper evaluation requires truly independent test sets.
- Figure 2 includes non-energy-conserving models like Orb without clear distinction in the plot. This skews the interpretation and unfairly impacts the perception of PET-MAD's strengths.
- The current benchmark results on Page 9 are particularly concerning as they arise directly from the training data split, making them statistically dependent and not meaningful for robust validation.

5. Overall Assessment

In summary, while the MAD dataset and PET-MAD model are useful contributions to the materials modeling community, the manuscript does not demonstrate sufficient advances in methodology, benchmarking rigor, or conceptual innovation to warrant publication in Nature Communications. The claims are overstated, the benchmarking is inadequate, and the novelty relative to existing literature is limited.

We therefore recommend rejection of this manuscript in its current form. The work could be better suited for a more specialized journal focused on computational materials science, where a more modest positioning of the contributions would be appropriate.

The manuscript "PET-MAD, a universal interatomic potential for advanced materials modeling" presents a pretrained universal machine learning interatomic potential (MLIP) trained on the massive atomistic diversity (MAD) dataset introduced in this paper. The authors claimed that they rival states-of-the-art MLIP based on their benchmarks and they also showcased the capabilities on six examples across various chemical and physical problems, and across various types of materials: inorganic, organic, surface, etc. The MAD dataset is indeed a good contribution to the research community, especially those who prefer Quantum Espresso + AiiDA. The PET-MAD model can also benefit some researchers. After reviewing this paper, we find that some of the benchmarks, and technical issues need to be addressed before it can be considered for publication in Nature Communications.

1. Benchmarks

In the past two years or so, the uMLIP community has gradually formed several popularly accepted benchmarks, including the matbench-discovery benchmark, the phonon prediction compared to phonondb dataset, the thermal conductivities predictions, and the MLIP-arena test suite. Although the listed benchmarks each may be biased, together they have already shown a wide range of capabilities. However, this manuscript claims to rival states-of-the-art performance without testing their model on any of the widely accepted metrics, which makes us to be skeptical about the benchmarks. Thus the claim of states-of-the-art performance is not solid.

In addition, the authors claimed 'rival' states-of-the-art performance in the abstract, what does 'rival' mean in this context? We don't think this is a scientific notation with a clear indication. A model can either be the states-of-the-art or not; if not, how much percentage worse than the states-of-the-art.

2. Questions about the presentation of the results

The results of the gamma phase show a strange comparison. For temperature lower than 625K, the bespoke version and the PET-MAD pretrained model have good agreements. This is ok. But, the fine-tuned version diverges from both bespoke and the PET-MAD models. Could you explain why this is happening?

In Fig. 6, how does your results compare with experimental measurements and first-principles MD results computed with PBE functional? Water has been extensively studied by many researchers and reported many times in literature. I'm interested in seeing if your bespoke model reproduced the PBE first-principles curve. Also, how does the distribution of the O-O-O angles in your trajectory?

One more point, the fine-tuning task is to tune the model parameters to map to the dataset of pure water at the same level of PBE functional theory, which is a comparatively easier task – as the model weights will just be shifted to make better prediction of a single system. However, if the PET-MAD model is really universal enough and its representation of the water system in the embedding space is expressive enough, the model should be able to fine-tune to datasets with higher level of theory, for example, in literature, the rev-PBE0-D3 has been adopted.

(Remarks on code availability)

Reviewer #2

(Remarks to the Author)

(Remarks on code availability)

Reviewer #3

(Remarks to the Author)

The manuscript submitted by Mazitov et al. presents PET-MAD, a machine learning interatomic potential (MLIP) based on the Point Edge Transformer (PET) architecture trained on the new Massive Atomistic Diversity (MAD) dataset. The authors aim to develop a "universal" potential that can accurately model diverse chemical systems including inorganic crystals, organic materials, surfaces, and clusters. The uncertainty quantification approach based on LLPR and shallow ensembles is a valuable addition that allows error propagation through complex workflows.

The fundamental approach is sound and the applications interesting, but in its current form, the manuscript does not represent the breakthrough advance that would warrant publication in Nature Communication.

I have several concerns about this work. The concept of a universal force field is not novel, as several such models already exist. The PET architecture itself was previously published, and combining diverse structure types in a training dataset (3D crystals, 2D materials, surfaces, molecules) has been done before, notably in the OpenLAM dataset that is benchmarked on the leaderboard of Matbench Discovery. The manuscript does not sufficiently clarify what fundamental advances this work makes beyond existing approaches. The authors train on a significantly smaller dataset (~95,000 structures) with fewer parameters than competing foundation models, but do not adequately explain why this approach should outperform more complex models with larger training sets.

The paper claims efficiency advantages, but independent testing done in my group suggests PET-MAD is approximately 200 times slower for geometry relaxation tasks compared to standard universal force fields on the MatBench leaderboard.

The benchmarking approach is not convincing:

- 1) The authors create ad hoc subsets of existing datasets rather than using established community benchmarks like MatBench Discovery.
- 2) The comparison between models trained on different DFT functionals (PBE vs. PBEsol) introduces systematic biases that are not properly addressed.
- 3) While the authors mention the limitations of the PBEsol functional, they don't explain why they selected this functional instead of PBE, making hard the comparison with other models on the leaderboard, neither they explore how this choice affects the model ability to reproduce experimental observations.
- 4) No direct comparison is made to experimental data for the six science cases, nor with other universal ML force fields, making it difficult to assess real-world applicability.

In conclusion, this work represents a valuable addition to the field of machine learning interatomic potentials, I recommend publication in a more specialized journal rather than Nature Communications. Prior to publication, the authors should:

1. Include standard benchmarks used by the community rather than custom subsets.
2. Provide a more thorough comparison with existing universal potentials on equal footing.
3. Address the computational efficiency concerns.
4. Clarify the specific advantages PET-MAD offers compared to existing universal potential.

(Remarks on code availability)

A member of my group tested the code for standard geometry optimization.

Reviewer #4

(Remarks to the Author)

The authors study a universal interatomic potential for modeling selected materials.

While I like the careful benchmarking efforts as the one presented, I feel they may take too much room in the main paper. It is reassuring but a bit boring to just see that the proposed model works competitively. I may be not the only reader who may wonder why this is the case. A discussion on this aspect will be helpful and should be added (after all the architecture used is not overly special or innovative...)

It is well-known that universal models can fail and be instable in molecular dynamics. I may have overlooked a statement by the authors, but it would be good to discuss this point clearly for the new model. In general limits of the approach would be important to document.

There are some refs that may be interesting for the authors. Recently Ganschka et al Sci Data 2025 have presented a large data set. Kabylda et al 2024 Arxiv provide a universal MLFF for molecules. Unke et al Sci Adv 2024 provides an MLFF for biomolecules.

Overall, I like the ms. It is an interesting effort and should ultimately be published in Nat Comms.

(Remarks on code availability)

Reviewer #5

(Remarks to the Author)

Machine-learning interatomic potentials (MLIPs) are machine-learning models that predict the forces acting on atoms from their local atomic environments, trained on quantum-mechanical reference calculations. The purpose of MLIPs is to accelerate by orders of magnitude atomistic simulations, such as molecular dynamics simulations.

Foundation-model/universal MLIPs have training sets that broadly cover the periodic table ("universal"). For universal MLIPs, the expectations are to enable "out-of-the-box" stable simulations that are at least qualitatively correct, and for fine-tuning on few reference calculations to result in an MLIP that enables quantitatively correct simulations.

The study develops a universal MLIP. Its contributions are:

- The MAD ("massive atomistic diversity") dataset with high chemical and structural diversity that covers the periodic table, calculated at a reasonable but not overly accurate level of DFT.
- A universal MLIP ("PET-MAD") trained on this dataset which fulfills the above expectations.

Essentially, the authors create a dataset, use an existing MLIP architecture to train a model on this dataset, and technically validate the resulting universal MLIP model. In particular, no new scientific question is solved or addressed with this model.

As such, the study will be of interest primarily for the MLIP community, for which it is of significant interest, and, to a lesser extent, for computational materials scientists and chemists generally. While this scope might be perceived as somewhat narrow for Nature Communications, similar studies have been accepted for publication before (e.g., DOIs 10.1038/s41467-019-10343-5, 10.1038/s41467-022-29939-5), and have been cited well.

The technical choices made in the study appear sound (e.g., stratified 80/10/10 train/test/validate splits, LoRA for fine-tuning, extensive validation covering different atomistic systems and properties) and meet the validation standards expected for universal MLIPs. The work supports the conclusions. The manuscript contains some overly simplifying or misleading statements, see my comments on the manuscript below.

Dataset

The MAD dataset contains 100k structures covering all relevant chemical elements. It is based on subsets and modified subsets of the Materials Cloud and the SHIFTML databases and contains bulk crystals, surfaces, clusters, 2d materials, molecular crystals, and neutral molecular fragments of molecular crystals. Subset modifications aim at compositional and structural diversity. Energies and forces are computed using DFT with the PBEsol functional.

- It would be helpful to show example structures from all the 8 subgroups described in Section 2.1, at least in the SI.
- In Section 2.1, the authors state to aim for "organic and inorganic" simulations, which they achieve by including inorganic crystals and molecular crystals. However, previous language in this section and in the abstract ("reliable for molecules") indicates applicability for molecules, but only (fragments of) molecular crystals were used to construct and validate the dataset (using the PBE variant for solids). Why weren't molecules (e.g., from QM9, PUBCHEM) included? If applicability to molecules was not in scope, the wording in the abstract and Section 2.1 should be made more precise.

MLIP

The introduced PET-MAD universal MLIP is an instance of an existing transformer-based graph neural network. It has 4 M parameters.

The authors validate their PET-MAD MLIP by

- comparing against four other universal MLIPs (MACE, MatterSim, Orb, SevenNet) on six datasets (MAD, MPtrj, Alexandria, SPICR, MD22, OC2020).
- reproducing six application scenarios:
 - ionic transport in lithium thiophosphate
 - melting point of gallium arsenide
 - surface element segregation in a high-entropy alloy
 - quantum effects in liquid water
 - nuclear magnetic resonance chemical shielding in alpha succinic acid
 - dielectric response in barium titanate

The PET-MAD model performs favorably compared to other models. The choice of competing universal MLIPs and datasets for the benchmark comparison is balanced and representative. Technical choices seem not to favour PET-MAD over other MLIPs (e.g., choice of small and large variants of these models). The application scenarios demonstrate good data efficiency

and performance of PET-MAD for a broad range of systems and properties.

Manuscript

- Introduction

The introduction ends with two paragraphs that describe the content of the study (MAD dataset, PET-MAD MLIP incl. validation). At this point, there are only implicit hints as to why MAD and PET-MAD would be interesting or useful for the general readership of Nature Communications. This usefulness should be briefly but explicitly stated.

- p.2: "For many years"

Please quantify "many" and/or provide a reference for the development of MLIPs.

- p.3: "the state-of-the-art PET graph neural network"

In which sense is this model demonstrably state-of-the-art?

- Section 2.1

The first sentence claims that previous universal MLIPs were trained on either inorganic crystals or molecules, citing refs. 14, 15, 16, 19. Before (p.2), refs. 11, 12, 13 were given for universal MLIPs. On what data were the universal MLIPs in refs. 11, 12, 13 trained? For example, the universal MACE-MP-0 (arXiv: 2401.00096) MLIP was trained on very diverse systems, including zeolites, metal-organic frameworks, aqueous interfaces, polymerisations, and many others. Please adjust and/or make the claim more precise.

- Section 2.2

The authors claim that the PET architecture "offers excellent flexibility and scaling to large datasets."

It is unclear what "flexibility" and "scaling" refer to exactly.

Please quantify this statement and provide evidence.

- Section 2.3

"...allow uncertainty quantification at no [computational] cost"

This is obviously false, as the only calculation that does not incur any computational cost is the empty calculation, i.e., no calculation. The corresponding statement in Section 4.6 is not wrong, but still imprecise.

(Remarks on code availability)

I briefly looked at the code, it seems reasonable, there are clear instructions in the README file.

Version 1:

Reviewer comments:

Reviewer #1

(Remarks to the Author)

Looks good. Green light on my side.

(Remarks on code availability)

Had a quick check but did not try to install. Overall looks good.

Reviewer #3

(Remarks to the Author)

The authors have made a clear and valuable effort to address the reviewers' requests comprehensively. They have substantially expanded upon the original paper by including significantly more detailed information, more data, more explanations. I am pleased with their reply and the additional material they have incorporated.

(Remarks on code availability)

everything fine with the code.

Reviewer #4

(Remarks to the Author)

The ms has improved a lot and is 'less boring' as it puts less emphasis on the benchmarking and more on insightful analyses in several materials usecases. I am pretty happy with the paper.

(Remarks on code availability)

Reviewer #5

(Remarks to the Author)

I find the authors' replies and manuscript improvements to the issues raised by all referees convincing.

The two following aspects seem particularly relevant:

- The PET-MAD model substantially improves on the Pareto frontier of training set size and predictive accuracy (Figure 1). As argued by the authors, data efficiency is an important consideration in the design and development of uMLIPs (and regular MLIPs, too), whose increased consideration in the community would allow shifting computational investments away from creating larger and larger training sets towards, e.g., increased agreement with experimental observations via reference data at higher levels of theory.

- The presented benchmarking actively ensures consistency in the reference calculations and clearly demonstrates the value of that (Figure S3). Inconsistency errors are bound to arise when mixing datasets without recalculation, and the authors' good practice can help improve current benchmarking practices for uMLIPs. The fact that not a single case of PET-MAD failure was observed over many non-trivial MD simulations (reply to Referee 4) is an encouraging sign and speaks of careful work on the authors' side.

I am satisfied with the authors replies to my comments and their changes to the manuscript, and recommend publication of the revised manuscript in Nature Communications.

(Remarks on code availability)

Overall response to reviewers

Given that Reviewers 1 and 2 wrote their reviews together, and repeated the same points several times - first individually and then jointly, we thought it would be good to highlight some of the key changes we made to make sure they do not get lost in the lengthy point-by-point response below.

Our objective with PET-MAD is to make a model that is suitable for *advanced materials modeling*, and that is *lightweight*, both in terms of computational cost and of the size of the dataset it relies on. We succeed to achieve these goals, as we demonstrate in the new Fig. 1, reproduced below. It shows the accuracy of various models on an external benchmark (a subset of the Matbench dataset) against their number of parameters, and the size of the dataset they are trained on.

PET-MAD is not the model with the lowest error (it doesn't aim to be!) but it shifts dramatically the Pareto front, achieving an accuracy competitive with several recent models while using a tiny fraction of the data.

A second point we want to emphasize is the need to assess the accuracy of the models in a *consistent* way. We use different DFT settings from those used in many previous datasets to improve convergence and remove some sources of inconsistency (such as the use of large Hubbard U corrections only for some structures). This means however that it is not possible to simply test PET-MAD against existing benchmark data, but it is necessary to re-compute some structures in a consistent way. We take great care to do so in our manuscript, considering also datasets that are extrapolative for both PET-MAD and other models (SPICE, MD22, OC2020). When tested consistently, PET-MAD outperforms much larger models such as ORB-v2 and MatterSim-5M. Similar results were observed in a recent preprint, where different MLIPs were compared in catalytic applications. We also perform many tests on realistic simulation scenarios that we verify against dedicated single-purpose models - again showing our dedication to a precise, quantitative assessment going well beyond blind benchmarks.

Third, PET-MAD is extremely feature-rich. It provides inexpensive ensemble-based uncertainty quantification, making it possible to estimate its accuracy on derived quantities (we demonstrate this for the melting point of GaAs, and we added an example on phonon dispersion curves). In this revision, we also added the ability to perform direct force predictions, which makes it as fast as the non-conservative ORB-v2 model. Given that we provide both direct and conservative predictions, this can be used to accelerate simulations without introducing artifacts. Even though both these

techniques have been introduced in separate (recent) publications, this is the first time they are made available in a universally-applicable machine-learning potential.

Referee: 1

Dr. Michele and coauthors present a new forcefield together with a training set which covers materials other than bulk systems. The work is solid. However, I did not find the work presenting any significantly novel ideas, neither does it show significant performance gain compared with SOTA. Therefore, I do not find it, in the current form at least, acceptable for Nat Comm but more like a fit to more specialized journals. However, I would still suggest giving it a shot - let authors rework on the organization of the paper to better present its novelty. I'm happy to see the modified version for any new insights. In addition to this, there are a few aspects that concern me quite a lot, see below:

We thank the Reviewer for the valuable feedback and suggestions. We have included the results of benchmarking the PET-MAD on Matbench Discovery and PhononDB in the revised version of the Manuscript. More details on that, as well as our point-by-point response to the Reviewers comments, can be found below.

1. Some claims are not true even in the abstract, say, "recent "universal" models deliver qualitative accuracy across the periodic table but are often biased toward low-energy configurations." This was true for early models like M3GNet and MACE-MP-0. Recent open datasets already cover non-local minima structure quite extensively, like META's OMat24. The related models perform really well under high-temperature regimes.

Even though we acknowledge the existence of advanced models trained on large datasets like OMat24, we note that none of them have been yet published in peer-reviewed literature. We selected models that are published and reasonably widely used to execute non-trivial simulation protocols. These models are trained on MPtrj and Alexandria - datasets which target mostly low-energy configurations. Still, we have rephrased our description to soften it as indeed these dataset may also contain structures that are not strictly equilibrium configurations, even though they are definitely less outlandish than some of the randomized configurations included in MAD.

2. "PET-MAD rivals state-of the-art MLIPs for inorganic solids, while also being reliable for molecules, organic materials, and surfaces." This is an overly generalized big claim. I agree that for some properties, PET-MAD might have achieved SOTA but definitely not for all of the metrics. Just to put here the test of the performance on MatBench Discovery's thermal conductivity. PET-MAD is certainly not the best (the authors can retest this for more reliable results).... I'm not saying that this benchmark is perfect or should it represent the overall performance of a universal forcefield. In any case, it is certainly one most popular benchmark. The current result clearly is a counterexample to "PET-MAD rivals state-of the-art MLIPs for inorganic solids" The authors need to be specific on such big claims, especially on the performance side...

Model ksrme

eSEN-30M-OAM 0.17

ORB v3 0.21

SevenNet-MF-ompa 0.317

GRACE-2L-OAM 0.294

eSEN-30M-MP 0.34

MACE-MPA-0 0.412

MatterSim v1 5M 0.574

DPA3-v2-OpenLAM 0.687

GRACE-1L-OAM 0.516
 SevenNet-I3i5 0.55
 MatRIS v0.5.0 MPtrj 0.861
 GRACE-2L-MPtrj 0.525
 DPA3-v2-MPtrj 0.959
 MACE-MP-0 0.647
 PET-MAD 1.244
 AlphaNet-MPtrj 1.31
 eqV2 M 1.771
 ORB v2 1.732
 eqV2 S DeNS 1.676
 ORB v2 MPtrj 1.725

We first would like to address the Reviewer’s concern about PET-MAD rivaling other state-of-the-art models. One of the key advantages of PET-MAD is its transferability in both organic and inorganic domains, which is granted by having the structures from both domains in the training set. Therefore, while benchmarking the model, we have naturally chosen both organic and inorganic datasets to compare our model against other popular universal MLIPs. As Table 1 in the Manuscript demonstrates (reproduced below), PET-MAD indeed rivals, and in fact outperforms other models in most of the cases - not only for its internal test set, but also for many of the external datasets.

Dataset	PET-MAD	MACE-MP-0-L	MatterSim-5M	Orb-v2	SevenNet
MAD	17.6 65.1	81.6 181.5	47.3 133.7	52.9 96.2	82.1 173.5
MPtrj	22.3 77.9	15.1 50.8	21.3 61.4	5.6 21.9	9.8 25.5
Matbench	31.3 —	58.5 —	38.2 —	37.9 —	47.5 —
Alexandria	49.0 66.8	65.4 79.5	21.2 39.9	13.2 10.5	47.6 70.3
OC2020	18.3 114.5	82.4 169.6	31.5 119.2	19.8 99.3	45.7 162.7
SPICE	3.7 59.5	10.6 166.8	21.3 145.6	59.0 140.8	11.3 139.1
MD22	1.9 65.6	9.4 182.9	28.6 160.4	174.3 220.7	11.1 146.2

Table 1. Comparison of PET-MAD accuracies on popular atomistic machine learning datasets against MACE-MP-0-L, MatterSim-5M, Orb-v2, and SevenNet-I3i5 models. For each dataset and model, mean absolute errors are reported for raw energy|forces prediction (in units of meV/atom|meV/Å). We don’t show force errors for the Matbench Discovery data, because they are not available in the reference data.

Cases in which PET-MAD delivers notably worse accuracy are typically related to the MPtrj and Alexandria datasets, which favor the models containing them in the training data: just as using the MAD test set unfairly benefits PET-MAD (as we acknowledge) using MPtrj and Alexandria as test sets unfairly benefits the models that include them in their training.

Next, we would like to discuss the Matbench Discovery benchmark. The main issue with this benchmark (and a reason why we didn’t include it in the original version of the Manuscript) is its *inconsistency* with PET-MAD’s baseline level of theory. This means that the leaderboard table reproduced by the Reviewer is meaningless, because it is impossible to separate the error of the model from the discrepancy between reference energetics.

In fact, as the Reviewer might have noticed, we put significant efforts in a careful and *consistent* assessment of the models in this work, in order to avoid mixing the actual error of the model with a discrepancy in underlying DFT data. In order to demonstrate how significant this contribution can

be, we added the results of benchmarking PET-MAD on a subset of the Matbench Discovery data in the new Supplementary Section S3. Our results show, that in a *naive* benchmarking of PET-MAD on Matbench Discovery data (WBM dataset) would yield an error in predicting the energies above hull around 138 meV/atom, while the difference in the DFT baselines can be as large as 120 meV/atom. However, a careful benchmarking of PET-MAD on a consistently recomputed subset of Matbench Discovery reduces the error to a reasonable value of 41 meV/atom, which is more than 3 times lower than the naively computed value. We believe that this example delivers a very important and clear message, that a certain revision of the standards in universal models' assessment is required in order to maintain usability of the benchmarks for a wider class of the models (i.e. not only for those trained on MaterialsProject-compatible datasets). Below, we provide a Figure with the benchmarking results for convenience.

Figure S3. Right panel: comparison of the energies above hull of a subset of structures from the WBM dataset, recomputed with DFT using MAD settings (MAD DFT, red dots), and predicted with PET-MAD (blue dots). The mean absolute error (MAE) in predicting the WBM values of the energy above hull in eV/atom is presented in the legend. This comparison demonstrates the upper limit of accuracy of any MAD-trained model on the Matbench-Discovery benchmark upon using non-consistent DFT settings, as the difference in baseline DFT energy between Matbench and MAD settings reaches 120 meV/atom. **Left panel:** comparison of the PET-MAD predictions of the energy above hull against the consistent reference, recomputed with MAD settings. The resulting error decreases from 138 to 41 meV/atom, revealing the actual accuracy of the model.

3. *Apart from smaller size, how is the dataset design principle different from the one from PFP? Towards universal neural network potential for material discovery applicable to arbitrary combination of 45 elements | Nature Communications This needs to be articulated better as the authors have already done surfaces. Molecules, bulk, disordered systems et al. and on a large scale.....*

We agree with the Reviewer, that the PFP dataset shares certain common features with the MAD dataset. Considering that, we have added a corresponding citation in the revised version of the Manuscript. However, we would like to note that there is a significant gap between covering 45 elements and covering 85 elements in the dataset (as we do in MAD), and more importantly - finding universal and consistent settings that yield a decent convergence rate of the DFT runs. Moreover, a dataset of 45 elements, while definitely being diverse, still cannot be considered as universal. These differences, in our opinion, highlight the novelty of this work.

4. *Figure 1 is not very easy to read... the colors are too close. The presentation needs to be changed. Also, it is not fair to compare in-distribution with other models' error where the test set is randomly chosen from the same pool of training set. The authors either should use some open benchmark or need to create a new benchmark that is independent of the training set distribution, aka, using a different rule to construct it. Examples could be: a. some bulk from Alex; b. some*

molecular data from MD17; 3. Some surface data from OC22; 4. Some off-local-minima data from MatterSim's codebase et al. Things like these reflect the performance better.

We thank the Reviewer for this suggestion, the color scheme of the Fig. 1 (now Fig. 2) was updated to be more readable. Regarding the next comment: *"The authors either should use some open benchmark or need to create a new benchmark that is independent of the training set distribution, aka, using a different rule to construct it. Examples could be: a. some bulk from Alex; b. some molecular data from MD17; 3. Some surface data from OC22; 4. Some off-local-minima data from MatterSim's codebase et al. Things like these reflect the performance better. - we note that this is exactly what we had done in Table 1 by comparing our accuracies on subsets of MPtrj, Alexandria, SPICE, MD22, OC20, Matbench Discovery (see Table 1). We would also like to pinpoint that following our consistency approach, this benchmarking exercise was performed on data, consistent with each models' training set DFT settings. We believe that this level of careful benchmarking is often overlooked in the universal MLIP community at present, and is another example of how our manuscript establishes better practices. We unfortunately struggle to understand what kind of improvements the Reviewer expected to see relative to this exercise, and can only hypothesize that they might have overlooked Table 1 as it came after the internal benchmarks. We have completely re-structured the benchmark section to begin with the external datasets, and then follow with the former Fig. 1 (now Fig. 2) to discuss the insights we can infer comparing the accuracy of the various model on the different sectors of MAD.*

5. In Fig.2, authors can safely remove non-conservative models like Orb. No need to compare until they get to the same level of performance to conservative models. Or use another way to present non-conserve models. The current presentation is a bit uncovering the contribution from this work as PET-MAD seems quite fast.

We thank the reviewer for their suggestion. Non-conservative models are now shown using dashed lines to distinguish them better from the conservative ones. We also presented a non-conservative head of the PET-MAD potential, which is as fast or faster than Orb-v2. We also included a discussion of the limitations of using non-conservative models during molecular dynamics (MD) (see Supplementary Section S10) and discuss how they can be used in combination with conservative forces using a multiple-time-step integrator. We note that even though this approach has been recently proposed by some of the authors, this is the first time it is made available and demonstrated for a universal potential – another example of how our manuscript brings novelty to this field.

6. PAGE 9 is very concerning. Again, the benchmarks should NEVER come from the construction of a training set, even if it is randomly split out. It is not a fair benchmark. Therefore, I highly doubt the current claims in these entire sections. New benchmarks need to be set up in statistically irrelevant manner compared with the training set.

We believe that we addressed Reviewer's concerns extensively in our answer to the Q.4. We now center the benchmark section on external benchmarks, and discuss Fig. 2 (former Fig. 1) as a measure of the accuracy of different models on different types of structures.

7. The dataset is constructed using PBEsol. Is there any particular reason for that? I initially thought this would be something like metaGGA or hybrid when I read the abstract. But it is not. The novelty needs to be clearly stated here then.... One point might be this is the first opensource dataset with molecules, surfaces, bulk et al with same level of theory, which is plausible. But this certainly is not enough for Nat Comms...

One of the points we are making is that in order to develop robust universal potentials it is essential to rely on *robust* reference calculations. Given that the materials project VASP-PBE setup lacks internal consistency (e.g. due to inconsistent use of Hubbard U corrections, and the use of spin-polarized calculations that can converge to different local minima) we decided to change it to more converged and consistent settings. We picked a setup that had been already checked carefully (in terms of accuracy of the pseudopotential calculations, etc. in the making of the MC3D dataset) and for which we could afford airtight convergence.

8. Consider giving a distribution of the elements of the training set. Some description on the number of atoms in the training set, the energy and force distribution will let readers get a better overall picture of the dataset.

We thank the Reviewer for their suggestion. We attach the figure showing the distribution of elements in the MAD dataset, as well as the data on energies and forces distribution below.

Figure XXX. Periodic table indicating the statistical representation of the elements present in the MAD dataset.

Figure XXX. Histograms of energy per atom (top) and force modulus (bottom) within the different subsets of the MAD dataset.

The dataset is already publicly available on <https://archive.materialscloud.org/record/2025.98>, and we are preparing a data publication describing it in detail.

Referee: 2

The manuscript “PET-MAD, a universal interatomic potential for advanced materials modeling” by Dr. Michele and coauthors presents PET-MAD, a machine learning interatomic potential (MLIP) trained on the newly introduced MAD dataset, which includes a wide range of materials beyond bulk crystalline solids, such as surfaces, nanoclusters, molecular crystals. The authors claim that PET-MAD rivals the performance of state-of-the-art MLIPs while providing broader material coverage, and they showcase several application examples.

While the MAD dataset and the PET-MAD model may offer useful resources for the community, after careful evaluation, we do not believe this manuscript meets the standards of novelty required for publication in Nature Communications. Below, I outline the major reasons for my recommendation to reject the manuscript.

We thank the Reviewer for the valuable comments. We outline our response to the following comments below.

1. Overstatement of Claims and Lack of Rigorous Benchmarking

Several claims made in the manuscript, including the abstract, are overstated or insufficiently supported. For example, the statement that “recent ‘universal’ models deliver only qualitative accuracy across the periodic table but are biased toward low-energy configurations” is no longer accurate for the latest generation of datasets and models (e.g., META’s OMat24).

No testing against community-accepted, rigorous benchmarks was performed, such as Matbench-Discovery (<https://matbench-discovery.materialsproject.org/>), PhononDB (see <https://github.com/atztogo/phonondb?tab=readme-ov-file> and the phonon benchmark paper: <https://arxiv.org/abs/2412.16551>), MLIP-Arena (<https://huggingface.co/spaces/atomind/mlip-arena>), or thermal conductivity datasets (developed in <https://arxiv.org/abs/2408.00755> and maintained as part of matbench-discovery). Instead, the benchmarks rely on splits from the training distribution, which does not provide a fair or meaningful assessment of the model's generalization capabilities. Without benchmarking on independent test sets and tasks, the manuscript's performance claims are not credible.

As we already outlined in our response to Reviewer 1, no models trained on OMat24 have been published in peer-reviewed literature yet. Although we do not question the quality of the OMat24 dataset, one of the main advantages that our PET-MAD model demonstrates is a possibility of training highly transferable models with *one thousandth* of the data in OMat24. This finding shows that an immense amount of compute time spent on creating very large datasets is not necessary.

We thank the Reviewer for a suggestion to do a more thorough benchmarking. We performed a careful and, what is even more important, a *consistent* analysis of the accuracy of PET-MAD on certain fractions of Matbench Discovery and PhononDB benchmarks, and discussed why the overall idea of a naive benchmarking of the universal MLIPs on inconsistent data may lead to totally misleading conclusions (see Supplementary Sections S3 and S8). We provided a figure with the results of PET-MAD in predicting the energy above hull on a Matbench Discovery benchmark, as well as a supporting discussion, in our response to Reviewer 1. We also attach below the results of PET-MAD in predicting the phonon band structures (including LLPR uncertainties) on a few structures from the PhononDB benchmark, for which we checked that the original PhononDB PBEsol and MAD-recomputed results are consistent (i.e. they have the RMSD of 3 cm^{-1} in the case of BeO).

Figure S7. LLPR ensembles of phonon bands (orange, thin lines) for three representative structures of the phononDB dataset predicted with PET-MAD, compared with reference DFT results (black, dashed lines). Average values of the bands are reported denoted by thick orange lines.

Regarding the following comment made by Reviewer: “*Instead, the benchmarks rely on splits from the training distribution, which does not provide a fair or meaningful assessment of the model's generalization capabilities. Without benchmarking on independent test sets and tasks, the manuscript's performance claims are not credible.*”, we would disagree. In fact, even in the original version of the Manuscript, we performed a thorough benchmarking of PET-MAD against other universal MLIPs on a few popular open-source benchmarks like MPtrj, Alexandria, OC2020, SPICE and MD22, which are not the part of the PET-MAD training set. In fact, MPtrj and Alexandria datasets were included in the training sets of a few reference models, which made this comparison even less “fair” for PET-MAD. We believe that our choice of starting with the internal

test might have been misleading, and for this reason we completely re-hauled the benchmark section to emphasize more the external datasets.

2. Insufficient Novelty in Dataset Construction

Although the MAD dataset extends across molecules, surfaces, and bulk systems, it is unclear how it meaningfully differs from or advances beyond existing datasets, such as those accompanying the “Towards universal neural network potential for material discovery” work published in Nat. Comm. Apart from the reduced size, the design principles and novelty compared to existing efforts are not sufficiently articulated.

Moreover, the choice of PBEsol as the computational level of theory is neither well-motivated nor clearly justified. Historically, the community has primarily relied on the PBE functional to generate large-scale datasets (e.g. Materials Project, Google’s Genome, Microsoft’s MatterSim, Meta’s OMAT), comprising hundreds of millions of data points, with a recent trend to shift to more accurate functionals such as r²SCAN (<https://arxiv.org/abs/2503.04070>). The authors should provide a clear rationale for selecting PBEsol.

First, as we already outlined in our response to Reviewer 1, the PFP dataset shares certain common features with the MAD dataset. Considering that, we have added a corresponding citation in the revised version of the Manuscript. However, we note that this dataset is still missing many elements of fundamental importance in materials science: Y, Zr, Os, Ga, Ge, As, Se. Compared to that, the introduced MAD dataset covers 85 elements across various system types. More importantly, it is computed using a consistent set of DFT settings, finding which and achieving a decent convergence rate of the baseline DFT calculations is a non-trivial task itself, and it becomes even more difficult with an increase in the number of elements covered. Regarding the Reviewer's comment on our choice of the exchange-correlation functional (PBEsol), given our goal of obtaining consistent energetics across a diverse dataset we chose a setup (functional, pseudopotential), which had already been thoroughly tested for the MC3D dataset, and then brought to airtight convergence for MAD dataset calculations. The choice of PBEsol in that case was motivated with slightly better accuracy for inorganic solids. We performed a few internal tests on MAD structures with the r²SCAN functional and observed a lower convergence rate, as well as a tendency to get stuck in local minima in the self-consistent cycle. Similar convergence issues were observed in the MatPES dataset paper. This reinforces our decision to base this first version on a well-established setup.

3. Technical Issues in the Results and Analysis

There are several inconsistencies and technical issues in the results presentation:

- The behavior of the fine-tuned PET-MAD model for the gamma phase at low temperatures (below 625 K) diverges unexpectedly from both the bespoke model and the pretrained PET-MAD model. This inconsistency is unexplained and raises concerns about the model’s fine-tuning robustness. Typically, the fine-tuned model should give similar or the same results as the bespoke model, but this is not the case in this figure.*
- In the water simulation example, no comparison is made to experimental data or first-principles MD simulations (e.g., using PBE). Without these comparisons, it is difficult to assess the physical reliability of the model for molecular systems. Water systems have been extensively studied, and experimental radial distribution functions and angle distributions (e.g., O–O–O angles) are well-known references. How closely does the fine-tuned PET-MAD model reproduce these established results?*
- Fine-tuning the PET-MAD model to a water dataset with the same level of theory is a relatively easy task. A truly universal model should be able to fine-tune toward higher-level datasets (e.g., revPBE0-D3), which would better demonstrate the expressiveness of the pretrained*

representations. For example, the authors may refer to the recent MatterTune manuscript (<https://arxiv.org/abs/2504.10655>).

We understand the concerns of the Reviewer. Although the conductivities for the LPS gamma phase at temperatures lower than 650K are very small: the models predict almost no ionic transport, which makes statistical convergence difficult. In any case, we retrained a fine-tuned potential which reached a better accuracy, and performed simulations for a longer time which eliminated the discrepancy. Figure 3 has been updated accordingly. Regarding the water (and other molecular) simulations, the results of the bespoke model are meant to represent the baseline PBEsol. For example, as our training curves for the bespoke PET model for water show (see Supplementary Section S10.5), the model achieves almost negligible error of 0.2 meV/atom compared to the reference DFT data. Therefore our comparison against the bespoke model essentially represents a comparison against PBEsol DFT. Comparing these results against experimental data, in our opinion, would introduce a methodological flaw, since by no means PET-MAD (or a bespoke PET model) can match the experiment data better than the underlying DFT theory allows. Regarding the comment that the Reviewer made on a fine-tuning to a higher level of theory (e.g. revPBE0-D3) - this is indeed one of the possible directions that can be explored in general. However, the purpose of all our fine-tuning exercises in this paper was to rather demonstrate that a pre-trained PET-MAD model can be safely used in many atomistic simulation scenarios, as its results match closely with the results of the bespoke models trained on the system-specific data. In other words, there is no need to fine-tune PET-MAD or create a bespoke model for every use case, since the base accuracy of the model against the reference DFT is far better than the intrinsic accuracy of DFT. Therefore, our results on water simulations, as well as NMR crystallography results for a succinic acid, demonstrate that an overall description of the molecular systems is captured properly if compared to the DFT data, even though DFT itself can show substantial deviations with respect to experiments.

4. Presentation Issues

Several figures require major improvements for clarity and fairness:

- *Figure 1 uses colors that are too similar and visually confusing. More importantly, comparing in-distribution test errors against other models trained and tested independently is misleading. Proper evaluation requires truly independent test sets.*
- *Figure 2 includes non-energy-conserving models like Orb without clear distinction in the plot. This skews the interpretation and unfairly impacts the perception of PET-MAD's strengths.*
- *The current benchmark results on Page 9 are particularly concerning as they arise directly from the training data split, making them statistically dependent and not meaningful for robust validation.*

We thank the Reviewer for their recommendations on the figures, which echoes those of Reviewer 1. We have updated the color scheme to make them easier to read and interpret. Additionally, we added the non-conservative direct forces prediction capabilities to PET-MAD in order to provide a fair inference speed comparison against the Orb-v2 model, where PET-MAD shows either the same or a better performance. We have also included a discussion on how the shortcomings of the non-conservative forces prediction during MD can be addressed using the multiple-time-step integration technique. As for the last comment: *"The current benchmark results on Page 9 are particularly concerning as they arise directly from the training data split, making them statistically dependent and not meaningful for robust validation."* - we again would like to bring the Reviewer's attention to the fact that we did both in-sample and out-of-sample benchmarking of the PET-MAD using the MPtrj, Alexandria, OC2020, MD22 and SPICE data, which were not included in the training set of our model, so no statistical bias took place even in the originally submitted version of the Manuscript.

5. Overall Assessment

In summary, while the MAD dataset and PET-MAD model are useful contributions to the materials modeling community, the manuscript does not demonstrate sufficient advances in methodology, benchmarking rigor, or conceptual innovation to warrant publication in Nature Communications. The claims are overstated, the benchmarking is inadequate, and the novelty relative to existing literature is limited.

We therefore recommend rejection of this manuscript in its current form. The work could be better suited for a more specialized journal focused on computational materials science, where a more modest positioning of the contributions would be appropriate.

As discussed above, it appears that - possibly because of lack of clarity in our presentation - this Reviewer had missed a whole set of external benchmarks, as well as six careful and challenging examples assessing the stability and accuracy of the model in the setting of complex, realistic simulation scenarios spanning different classes of materials, static and dynamical properties, modeling of quantum nuclear effects, and the calculation of challenging properties such as nuclear magnetic resonance shielding values and temperature-dependent dielectric constants. We hope that with the revised presentation, the additional benchmarks on phonons and the Matbench dataset, and the inclusion of a hybrid conservative/non-conservative simulation that accelerates simulations twofold without breaking energy conservation, they will be less dismissive of the value of the present manuscript.

Referee: 1 + 2 (joint)

The manuscript "PET-MAD, a universal interatomic potential for advanced materials modeling" presents a pretrained universal machine learning interatomic potential (MLIP) trained on the massive atomistic diversity (MAD) dataset introduced in this paper. The authors claimed that they rival states-of-the-art MLIP based on their benchmarks and they also showcased the capabilities on six examples across various chemical and physical problems, and across various types of materials: inorganic, organic, surface, etc. The MAD dataset is indeed a good contribution to the research community, especially those who prefer Quantum Espresso + AiiDA. The PET-MAD model can also benefit some researchers. After reviewing this paper, we find that some of the benchmarks, and technical issues need to be addressed before it can be considered for publication in Nature Communications.

We thank Reviewers 1 and 2 for their thorough analysis of our work and their valuable suggestions. We believe these suggestions will make our statements more solid and convincing. Some of the questions raised in the Joint Review were already answered in the individual responses to Reviewers 1 and 2, so we will address the reader to these sections of the Response Letter in the case of overlap. Here, we address the final Reviewers' concerns and emphasize a few aspects of our work that could have been misinterpreted.

1. Benchmarks

In the past two years or so, the uMLIP community has gradually formed several popularly accepted benchmarks, including the matbench-discovery benchmark, the phonon prediction compared to phonondb dataset, the thermal conductivities predictions, and the MLIP-arena test suite. Although the listed benchmarks each may be biased, together they have already shown a wide range of capabilities. However, this manuscript claims to rival states-of-the-art performance without testing

their model on any the widely accepted metrics, which makes us to be skeptical about the benchmarks. Thus the claim of states-of-the-art performance is not solid.

In addition, the authors claimed ‘rival’ states-of-the-art performance in the abstract, what does ‘rival’ mean in this context? We don’t think this is a scientific notation with a clear indication. A model can either be the states-of-the-art or not; if not, how much percentage worse than the states-of-the-art.

As discussed in the individual responses to Reviewers 1 and 2, we included the benchmarking results of the PET-MAD model on certain fractions of Matbench Discovery and PhononDB benchmarks. We tried to deliver a clear message, that a naive assessment of the model trained on data, which is internally inconsistent with the benchmark data, would yield misleading results, where the difference in the underlying DFT reference contributes significantly to the total error. Therefore, overall applicability of such benchmarks is questionable for any model not trained on the data consistent with Materials Project settings. Considering that, in the original version of the Manuscript, we included a few benchmarks that cover various potential use-cases of the model in both inorganic and organic domains, accounting for DFT consistency for each reference model. We believe that this kind of benchmarking approach is much more comprehensive and thus more valuable.

Dataset	PET-MAD	MACE-MP-0-L	MatterSim-5M	Orb-v2	SevenNet
MAD	17.6 65.1	81.6 181.5	47.3 133.7	52.9 96.2	82.1 173.5
MPtrj	22.3 77.9	15.1 50.8	21.3 61.4	5.6 21.9	9.8 25.5
Matbench	31.3 —	58.5 —	38.2 —	37.9 —	47.5 —
Alexandria	49.0 66.8	65.4 79.5	21.2 39.9	13.2 10.5	47.6 70.3
OC2020	18.3 114.5	82.4 169.6	31.5 119.2	19.8 99.3	45.7 162.7
SPICE	3.7 59.5	10.6 166.8	21.3 145.6	59.0 140.8	11.3 139.1
MD22	1.9 65.6	9.4 182.9	28.6 160.4	174.3 220.7	11.1 146.2

Table 1. Comparison of PET-MAD accuracies on popular atomistic machine learning datasets against MACE-MP-0-L, MatterSim-5M, Orb-v2, and SevenNet-I3i5 models. For each dataset and model, mean absolute errors are reported for raw energy|forces prediction (in units of meV/atom|meV/°A). We don’t show force errors for the Matbench Discovery data, because they are not available in the reference data.

Since we claimed to push the limits of “universality”, we should have assessed PET-MAD against other models using this extended set of benchmarks, in which PET-MAD indeed delivers better accuracy in most of the cases (Table 1, reproduced above). This is why we claim that our model *rivals* state-of-the-art universal MLIPs. Not only PET-MAD delivers better accuracy in general, it also outperforms almost all reference models in terms of speed (and shows similar or better inference timings compared to the Orb-v2 model in the non-conservative regime see Fig. 3, reproduced below).

Figure 3. Inference time of several models evaluated over different bulk materials and system sizes. For each MLIP, we use its LAMMPS interface if available, preferably choosing the Kokkos-enabled (kk) version, or its ASE interface otherwise. All timings were measured on a single NVIDIA H100 GPU. Missing points for MACE-MP-0 and SevenNet caused by out-of-memory errors are marked with red crosses. The non-conservative (NC) versions of Orb-v2 and PET-MAD are shown in dashed lines. These models benefit from a theoretical speedup, but can violate the conservation of energy, often resulting in ill-behaved molecular dynamics. The model versions used were MatterSim-v1.0.0-1M, MACE-MP-0 (M), Orb-v2, SevenNet-0 (11Jul2024).

This is another highly important feature of our model, which is sometimes even more valuable for users working with molecular dynamics simulations. We also would like to emphasize that all these results were achieved by training the model on the dataset, which is 1-3 orders of magnitude smaller than other dataset typically used for creation of universal MLIPs, and has a relatively small number of trainable parameters. We believe that data-efficiency is another quite important property of the models, which is often overlooked in the community, and can potentially save huge amounts of computational resources if considered seriously.

Last, but not the least, we offer a convenient user interface for PET-MAD implemented in a few open-source simulation packages, like ASE, LAMMPS (with a full GPU support) and PLUMED, supported with extensive documentation, user tutorials on installation, running the simulations, performing the uncertainty quantification and fine-tuning PET-MAD, which certainly serves as another valuable contribution to the field. All these features are building blocks of our claim that PET-MAD is competitive with other state-of-the-art universal models, and we believe all of them are important enough to be considered jointly when assessing the quality of our work.

2. Questions about the presentation of the results

The results of the gamma phase show a strange comparison. For temperature lower than 625K, the bespoke version and the PET-MAD pretrained model have good agreements. This is ok. But, the fine-tuned version diverges from both bespoke and the PET-MAD models. Could you explain why this is happening?

In Fig. 6, how does your results compare with experimental measurements and first-principles MD results computed with PBE functional? Water has been extensively studied by many researchers and reported many times in literature. I'm interested in seeing if your bespoke model reproduced the PBE first-principles curve. Also, how does the distribution of the O-O-O angles in your trajectory?

One more point, the fine-tuning task is to tune the model parameters to map to the dataset of pure water at the same level of PBE functional theory, which is a comparatively easier task – as the model weights will just be shifted to make better prediction of a single system. However, if the PET-MAD model is really universal enough and its representation of the water system in the

embedding space is expressive enough, the model should be able to fine-tuned to datasets with higher level of theory, for example, in literature, the rev-PBE0-D3 has been adopted.

We thank the Reviewers for their comments and suggestions regarding the presentation of our results. Since the content of these concerns was already extensively covered in our response to Reviewer 2, we address the reader to the p. 3 of the corresponding response.

Referee: 2 (Remarks to the Author)

Referee: 3

The manuscript submitted by Mazitov et al. presents PET-MAD, a machine learning interatomic potential (MLIP) based on the Point Edge Transformer (PET) architecture trained on the new Massive Atomistic Diversity (MAD) dataset. The authors aim to develop a "universal" potential that can accurately model diverse chemical systems including inorganic crystals, organic materials, surfaces, and clusters. The uncertainty quantification approach based on LLPR and shallow ensembles is a valuable addition that allows error propagation through complex workflows.

The fundamental approach is sound and the applications interesting, but in its current form, the manuscript does not represent the breakthrough advance that would warrant publication in Nature Communication.

I have several concerns about this work. The concept of a universal force field is not novel, as several such models already exist. The PET architecture itself was previously published, and combining diverse structure types in a training dataset (3D crystals, 2D materials, surfaces, molecules) has been done before, notably in the OpenLAM dataset that is benchmarked on the leaderboard of Matbench Discovery. The manuscript does not sufficiently clarify what fundamental advances this work makes beyond existing approaches. The authors train on a significantly smaller dataset (~95,000 structures) with fewer parameters than competing foundation models, but do not adequately explain why this approach should outperform more complex models with larger training sets.

We thank the Reviewer for constructive comments and careful evaluation of our work. While the development of universal MLIPs is indeed not a new idea, we believe that PET-MAD has a few important advantages, which contribute to the field in several novel and impactful ways. As the Reviewer notes, one key difference in PET-MAD is that it breaks with the trend of increasing model and dataset size, and shows that with a clever selection of the training structures and model architecture it is possible to shift substantially the Pareto front, achieving an accuracy that is similar or even better than models trained on up to 1000 times more structures (see table below)

Dataset	PET-MAD	MACE-MP-0-L	MatterSim-5M	Orb-v2	SevenNet
MAD	17.6 65.1	81.6 181.5	47.3 133.7	52.9 96.2	82.1 173.5
MPtrj	22.3 77.9	15.1 50.8	21.3 61.4	5.6 21.9	9.8 25.5
Matbench	31.3 —	58.5 —	38.2 —	37.9 —	47.5 —
Alexandria	49.0 66.8	65.4 79.5	21.2 39.9	13.2 10.5	47.6 70.3
OC2020	18.3 114.5	82.4 169.6	31.5 119.2	19.8 99.3	45.7 162.7
SPICE	3.7 59.5	10.6 166.8	21.3 145.6	59.0 140.8	11.3 139.1
MD22	1.9 65.6	9.4 182.9	28.6 160.4	174.3 220.7	11.1 146.2

Table 1. Comparison of PET-MAD accuracies on popular atomistic machine learning datasets against MACE-MP-0-L, MatterSim-5M, Orb-v2, and SevenNet-I3i5 models. For each dataset and model, mean absolute errors are reported for raw energy|forces prediction (in units of meV/atom|meV/°A). We don't show force errors for the Matbench Discovery data, because they are not available in the reference data.

This is an extremely important message for the field, as in our opinion it suggests the opportunity of allocating computational resources in a different direction, e.g. by increasing the level of theory at which structures are evaluated. The choice of a lightweight architecture also makes the model *fast* (more on that below) which is important when running advanced simulation workflows. Incidentally, we also demonstrate a further 2x speedup by combining conservative and non-conservative force predictions, a recently-developed idea that we apply here to a universal forcefield for the first time.

We also note that this is a very different philosophy from OpenLAM (which incidentally was published less than a month before we submitted this paper) that concatenates datasets with different origins, and computed with inconsistent DFT settings, as it emphasizes the importance of internal consistency with the dataset. We had already paid great attention to internal consistency and to assess model accuracy against a consistent target. In this revision, we also provide an extensive discussion of the importance of consistency, especially when performing benchmarks, demonstrating how the differences in DFT settings can lead to differences in energy differences and forces that far exceed the accuracy of models against their consistent reference energetics - making the benchmarks completely meaningless.

The paper claims efficiency advantages, but independent testing done in my group suggests PET-MAD is approximately 200 times slower for geometry relaxation tasks compared to standard universal force fields on the MatBench leaderboard.

Regarding the Reviewer's concerns about the geometry optimization speed, we performed a geometry optimization test based on structures from the PhononDB benchmark, and included these results to the SI (see Supplementary Section S6). On average, PET-MAD demonstrates the best overall geometry optimization speed and robustness, compared to other considered reference models (see Figure below). Orb-v2, being non-conservative, has faster inference while being less stable, as we show in the number of failed optimizer runs. We obviously cannot guess what went wrong in the benchmarks performed by the Reviewer's collaborators, and we are clearly concerned that - despite our efforts to provide an easy-to-use implementation - incorrect settings could lead to such dramatic degradation in performance. We invite the Reviewers' collaborator to open an issue in the PET-MAD GitHub repository (<https://github.com/lab-cosmo/pet-mad>) if the problem persists, so that we can make sure that PET-MAD performs as it should in every possible setup.

Figure S5. Timing distributions for 1000 geometry optimization runs using the LBFGS optimizer. The black dashed band represents PET-MAD timings within three σ from the median under the Gaussian approximation (0.135th to 99.865th percentile). The orange circles are the number of times the optimization runs did not complete.

Below, we provide point-to-point answers to Reviewer's comments.

The benchmarking approach is not convincing:

1) *The authors create ad hoc subsets of existing datasets rather than using established community benchmarks like MatBench Discovery.*

As we discussed above, we were already performing several benchmarks against external datasets (many of which are part of the OpenLAM dataset, that the Reviewer has mentioned). In doing so, we had made sure to recompute a representative subset with parameters consistent with the training target of each dataset. We also include benchmarking results on an appropriately recomputed subset of Matbench Discovery, that shows PET-MAD has competitive performance despite the much smaller training set. We believe that we provided a thorough explanation of why we had to use the recomputed subsets, including the case of the Matbench Discovery benchmark.

2) *The comparison between models trained on different DFT functionals (PBE vs. PBEsol) introduces systematic biases that are not properly addressed.*

We agree entirely that comparing models trained with different DFT settings against inconsistent datasets make benchmarks meaningless, but we disagree that we did not address this properly. Precisely to avoid these biases, we took care of recomputing a representative subset with consistent DFT settings. Every model that we considered as a reference was benchmarked against its own version of the benchmarking subset, recomputed consistently with the data that the model

was trained on. Therefore, PET-MAD was compared against the data computed with PBEsol, and other models - against PBE data, recomputed consistently with the Materials Project datasets settings (which MPTrj and Alexandria datasets are based on).

3) *While the authors mention the limitations of the PBEsol functional, they don't explain why they selected this functional instead of PBE, making hard the comparison with other models on the leaderboard, neither they explore how this choice affects the model ability to reproduce experimental observations.*

We have already discussed above that benchmarks like Matbench Discovery have the issue of being internally inconsistent - e.g. having a large Hubbard U correction applied depending on the composition of a given structure. We claim that accurate MLIPs require highly consistent underlying data in order to achieve good, transferable accuracy. To see why doing otherwise would be a problem the Reviewer can consider (or realize in practice) an example of an interface between a transition metal oxide (treated with U corrections) and a transition metal (treated without). Therefore, we selected a set of settings, which is consistent and robust across different systems with different elements. Having to pick different DFT settings, and since we would in any case break with "tradition", we decided to choose settings that had been thoroughly tested for consistency and SCF stability in the making of the MC3D dataset, which forms one part of MAD. This dataset used PBEsol, which often yields slightly better accuracy against experiments than PBE for inorganic crystals, and we considered this to be an equally valid choice for our purposes.

4) *No direct comparison is made to experimental data for the six science cases, nor with other universal ML force fields, making it difficult to assess real-world applicability.*

The six science cases are meant to test quantitatively the accuracy of PET-MAD when following complicated simulation protocols. As such, the only meaningful target for comparison is the DFT the model is trained on (as obviously PET-MAD cannot match the experiment data better than the underlying DFT theory allows). Comparison with other universal ML forcefields would be equally meaningless: if there was a discrepancy, how could we tell which model is correct? This is the reason why we chose examples for which single-purpose training sets were available, and we recomputed them with consistent DFT settings. This allowed us to train bespoke and fine-tuned models that match the target DFT to a higher degree than PET-MAD, and serve as a proxy for a DFT reference (that would be, for all our examples, prohibitively expensive).

In conclusion, this work represents a valuable addition to the field of machine learning interatomic potentials, I recommend publication in a more specialized journal rather than Nature Communications. Prior to publication, the authors should:

1. *Include standard benchmarks used by the community rather than custom subsets.*
2. *Provide a more thorough comparison with existing universal potentials on equal footing.*
3. *Address the computational efficiency concerns.*
4. *Clarify the specific advantages PET-MAD offers compared to existing universal potential.*

We explain above how

1. We do test against standard benchmarks (MPTrj, Alexandria, Matbench, SPICE, OC2020, MD22) and the only reason we select a representative subset is to ensure consistent energetics and therefore a meaningful comparison - as doing otherwise will lead to unacceptable biases, as noted by this Reviewer
2. We do compare accuracy and speed with 4 widely used universal forcefields, and we have now included an even more thorough comparison using Matbench

3. We have demonstrated that the speed of geometry optimization is consistent with the MD benchmarks we had reported. Whatever the issue the Reviewer has encountered, we cannot reproduce without further details and we recommend filing a GitHub issue to resolve it.
4. Compared to the four models we compare with, PET-MAD is faster, more accurate on several external datasets, and is trained on a much smaller dataset which makes it easier to re-train, and to upgrade to higher levels of electronic structure theory.

Given we have responded thoroughly to all the criticism of this Reviewer, we hope they will update their recommendation accordingly.

Referee: 4

The authors study a universal interatomic potential for modeling selected materials.

While I like the careful benchmarking efforts as the one presented, I feel they may take too much room in the main paper. It is reassuring but a bit boring to just see that the proposed model works competitively. I may be not the only reader who may wonder why this is the case. A discussion on this aspect will be helpful and should be added (after all the architecture used is not overly special or innovative...)

We thank the Reviewer for valuable feedback and recommendations. We are also not especially thrilled by benchmarks - which is why we had also included six examples of usage in complex MD setups, including simulations with quantum nuclear statistics, melting point calculations, uncertainty quantification and (in this revised version) hybrid conservative/non-conservative multiple time stepping integration.

Nevertheless, given the criticism from other Reviewers who deemed the benchmarks insufficient, we believe that a careful analysis of the benchmarks is necessary to support our claims in this work. As the Reviewer might have noticed, we put significant efforts trying to pursue the most appropriate way of analysing the models' accuracies (both PET-MAD and the reference models), which requires comparing the models' predictions against the reference data, computed consistently with the corresponding training sets. The price of not doing that is a lack of ability to distinguish the models' errors from the errors that come from the underlying discrepancies in the DFT data. While preparing the updated version of the manuscript, we have added a new benchmark of PET-MAD on Matbench Discovery data (see Supplementary Section S3), where we show that these discrepancies can reach 120 meV/atom, which is a 3 times larger error compared to the actual accuracy of PET-MAD, when performed consistently - i.e. 41 meV/atom. We added a discussion of these findings in hopes of delivering a clearer message about why this is so important.

It is well-known that universal models can fail and be instable in molecular dynamics. I may have overlooked a statement by the authors, but it would be good to discuss this point clearly for the new model. In general limits of the approach would be important to document.

As the Reviewer mentioned, performing the molecular dynamics (MD) with universal MLIPs indeed can fail, and this is what we have seen with other reference models from time to time. However, while preparing this paper, we did not observe a single case of PET-MAD failing on MD in all the case-studies, which include rather complex simulation protocols. As a further safeguard, PET-MAD provides built-in (and inexpensive) uncertainty quantification, both for individual predictions and (through ensemble propagation) for the final observables. We believe that the stability in all of our

six, diverse case studies and the presence of error estimates provide a solid foundation for the safe use of PET-MAD in simulation studies.

There are some refs that may be interesting for the authors. Recently Ganscha et al Sci Data 2025 have presented a large data set. Kabylda et al 2024 Arxiv provide a universal MLFF for molecules. Unke et al Sci Adv 2024 provides an MLFF for biomolecules.

We thank the Reviewer for sharing the links to other universal MLIPs for molecular studies. While we have definitely found them quite useful (and, in fact, the MD22 benchmark we use was inspired by one of these works), we unfortunately had difficulty including them in the list of references due to the strict (and in our opinion misguided) limitations on the number of references coming from the Nature Communications journal policy.

Overall, I like the ms. It is an interesting effort and should ultimately be published in Nat Comms.

We thank the Reviewer for their supportive recommendation.

Referee: 5

Machine-learning interatomic potentials (MLIPs) are machine-learning models that predict the forces acting on atoms from their local atomic environments, trained on quantum-mechanical reference calculations. The purpose of MLIPs is to accelerate by orders of magnitude atomistic simulations, such as molecular dynamics simulations.

Foundation-model/universal MLIPs have training sets that broadly cover the periodic table ("universal"). For universal MLIPs, the expectations are to enable "out-of-the-box" stable simulations that are at least qualitatively correct, and for fine-tuning on few reference calculations to result in an MLIP that enables quantitatively correct simulations.

The study develops a universal MLIP. Its contributions are:

- The MAD ("massive atomistic diversity") dataset with high chemical and structural diversity that covers the periodic table, calculated at a reasonable but not overly accurate level of DFT.*
- A universal MLIP ("PET-MAD") trained on this dataset which fulfills the above expectations.*

Essentially, the authors create a dataset, use an existing MLIP architecture to train a model on this dataset, and technically validate the resulting universal MLIP model. In particular, no new scientific question is solved or addressed with this model.

As such, the study will be of interest primarily for the MLIP community, for which it is of significant interest, and, to a lesser extent, for computational materials scientists and chemists generally. While this scope might be perceived as somewhat narrow for Nature Communications, similar studies have been accepted for publication before (e.g., DOIs 10.1038/s41467-019-10343-5, 10.1038/s41467-022-29939-5), and have been cited well.

The technical choices made in the study appear sound (e.g., stratified 80/10/10 train/test/validate splits, LoRA for fine-tuning, extensive validation covering different atomistic systems and properties) and meet the validation standards expected for universal MLIPs. The work supports the conclusions. The manuscript contains some overly simplifying or misleading statements, see my comments on the manuscript below.

We thank the Reviewer for such a high assessment of our work, as well as for a thorough analysis, and valuable recommendations. Below, we provide our point-to-point response to their questions and comments.

Dataset

The MAD dataset contains 100k structures covering all relevant chemical elements. It is based on subsets and modified subsets of the Materials Cloud and the SHIFTML databases and contains bulk crystals, surfaces, clusters, 2d materials, molecular crystals, and neutral molecular fragments of molecular crystals. Subset modifications aim at compositional and structural diversity. Energies and forces are computed using DFT with the PBEsol functional.

- It would be helpful to show example structures from all the 8 subgroups described in Section 2.1, at least in the SI.

We agree with the Reviewer that this paper lacks a thorough description of the MAD dataset. This was done on purpose in order to keep the focus of this paper on the model and its features. Regarding the MAD dataset - we intended to prepare a separate paper with a more extensive discussion and visualizations. The current version of that paper can be found on arXiv: <https://doi.org/10.48550/arXiv.2506.19674>.

- In Section 2.1, the authors state to aim for "organic and inorganic" simulations, which they achieve by including inorganic crystals and molecular crystals. However, previous language in this section and in the abstract ("reliable for molecules") indicates applicability for molecules, but only (fragments of) molecular crystals were used to construct and validate the dataset (using the PBE variant for solids). Why weren't molecules (e.g., from QM9, PUBCHEM) included? If applicability to molecules was not in scope, the wording in the abstract and Section 2.1 should be made more precise.

Even though the molecular datasets mentioned by the Reviewer are indeed widely used in the community, we decided to stay with our minimalistic description of molecular systems in the MAD dataset for a few following reasons. First, we tried to build our dataset with a philosophy of a gradual increase in structural diversity, thus a transition from molecular crystals to derivative molecular fragments was a natural choice. As we demonstrate in our benchmarks, having these fragments (also accompanied with nanoclusters), yields a decent accuracy for SPICE and MD22 datasets, which cover much larger space of molecular systems, that includes amino acids, fatty acids, nucleic acids, as well as the PubChem molecules (as a part of the SPICE dataset). We believe that this approach, while for sure leaving space for future improvements, allowed us to stay minimalistic and data-efficient in terms of the overall dataset size.

MLIP

The introduced PET-MAD universal MLIP is an instance of an existing transformer-based graph neural network. It has 4 M parameters.

The authors validate their PET-MAD MLIP by

- comparing against four other universal MLIPs (MACE, MatterSim, Orb, SevenNet) on six datasets (MAD, MPtrj, Alexandria, SPICR, MD22, OC2020.

- reproducing six application scenarios:

ionic transport in lithium thiophosphate
 melting point of gallium arsenide
 surface element segregation in a high-entropy alloy
 quantum effects in liquid water
 nuclear magnetic resonance chemical shielding in alpha succinic acid
 dielectric response in barium titanate

The PET-MAD model performs favorably compared to other models. The choice of competing universal MLIPs and datasets for the benchmark comparison is balanced and representative. Technical choices seem not to favour PET-MAD over other MLIPs (e.g., choice of small and large variants of these models). The application scenarios demonstrate good data efficiency and performance of PET-MAD for a broad range of systems and properties.

Manuscript

- Introduction

The introduction ends with two paragraphs that describe the content of the study (MAD dataset, PET-MAD MLIP incl. validation). At this point, there are only implicit hints as to why MAD and PET-MAD would be interesting or useful for the general readership of Nature Communications. This usefulness should be briefly but explicitly stated.

This is an excellent suggestion. We now conclude the introduction with a one-sentence summary:

PET-MAD demonstrates that it is possible to train universal models that are accurate, fast, and robust using a tiny fraction of the structures of last-generation datasets, and provides a feature-rich framework for advanced atomistic simulations, including direct-force acceleration and inexpensive end-to-end uncertainty quantification.

- p.2: "For many years"

Please quantify "many" and/or provide a reference for the development of MLIPs.

We changed the expression to "For almost two decades", as the first papers proposing "modern" MLIPs appeared around 2005-2007 (Behler and Parrinello, PRL 2007 is a typical reference). Given the fact that Nature Communications limits the total number of references, and to avoid giving a too narrow perspective on the early developments of the field, we prefer to cite some recent reviews that provide a broader discussion to the interested reader.

- p.3: "the state-of-the-art PET graph neural network"

In which sense is this model demonstrably state-of-the-art?

At the time of its publication [Pozdnyakov, Ceriotti. Advances in NIPS (2023)] the PET architecture demonstrated SOTA performance compared to other architectures. We do acknowledge however that this language is imprecise and unnecessary, so we removed it and only refer to PET as a rotationally unconstrained GNN architecture.

- Section 2.1

The first sentence claims that previous universal MLIPs were trained on either inorganic crystals or molecules, citing refs. 14, 15, 16, 19. Before (p.2), refs. 11, 12, 13 were given for universal MLIPs. On what data were the universal MLIPs in refs. 11, 12, 13 trained? For example, the universal

MACE-MP-0 (arXiv: 2401.00096) MLIP was trained on very diverse systems, including zeolites, metal-organic frameworks, aqueous interfaces, polymerisations, and many others. Please adjust and/or make the claim more precise.

According to the original paper of Batatia et al., MACE-MP-0 was trained on MPtrj dataset (technically, this is why it has “MP” in its name, which refers to Materials Project as a source of training data). However, the model was indeed *tested* on different benchmarks, that include zeolites, MOFs, etc. – although, crucially in our opinion, mostly in a qualitative way, without performing the quantitative assessment against a bespoke model with the same DFT target as we do here. Similarly, refs. 11, 12, 13 are also trained on datasets that predominantly consist of inorganic solids - although they do contain some small molecular moieties such as water and simple organic compounds, giving them some ability to extrapolate to organic systems - but with low accuracy as evidenced by the large errors on SPICE and MD22, cf. Table 1.

That said, we think this sentence might have been misleading, as our intention was to emphasize that previous *datasets* usually focused on either organic or inorganic structures. We have changed the opening of Sec. 2.1 to read

Most of the existing efforts to generate datasets to train universal models focus on either inorganic crystals~\cite{Deng2023, Schmidt2023, Wang2023} or molecular compounds~\cite{east+23sd}, and aim to include as many structures as possible.

- Section 2.2

The authors claim that the PET architecture "offers excellent flexibility and scaling to large datasets."

It is unclear what "flexibility" and "scaling" refer to exactly.

Please quantify this statement and provide evidence.

We again thank the Reviewer for calling out imprecise and generic language. We replaced it with the following statement:

[...] has a high descriptive power (as every transformer layer can be scaled to be a universal approximator)\cite{pozd-ceri23nips} and low inference cost (as we shall demonstrate in Fig.~\ref{fig:speed}).

- Section 2.3

"...allow uncertainty quantification at no [computational] cost"

This is obviously false, as the only calculation that does not incur any computational cost is the empty calculation, i.e., no calculation. The corresponding statement in Section 4.6 is not wrong, but still imprecise.

We thank the Reviewer for this recommendation. We intended to say that in the case of PET-MAD, the uncertainty quantification can be done with *negligible additional* cost, as the shallow-ensemble architecture it uses involves just an ensemble of linear last layers rather than several full re-evaluations of the computationally demanding part of the model. We clarified this point in the text.